# Constrained Belief Updates Explain Geometric Structures in Transformer Representations

**Mateusz Piotrowski** [1]   **Paul M. Riechers** [2 3]   **Daniel Filan** [1]   **Adam S. Shai** [2]

## Abstract

What computational structures emerge in transformers trained on next-token prediction? In this work, we provide evidence that transformers implement constrained Bayesian belief updating—a parallelized version of partial Bayesian inference shaped by architectural constraints. We integrate the model-agnostic theory of optimal prediction with mechanistic interpretability to analyze transformers trained on a tractable family of hidden Markov models that generate rich geometric patterns in neural activations. Our primary analysis focuses on single-layer transformers, revealing how the first attention layer implements these constrained updates, with extensions to multi-layer architectures demonstrating how subsequent layers refine these representations. We find that attention carries out an algorithm with a natural interpretation in the probability simplex, and create representations with distinctive geometric structure. We show how both the algorithmic behavior and the underlying geometry of these representations can be theoretically predicted in detail—including the attention pattern, OV-vectors, and embedding vectors—by modifying the equations for optimal future token predictions to account for the architectural constraints of attention. Our approach provides a principled lens on how architectural constraints shape the implementation of optimal prediction, revealing why transformers develop specific intermediate geometric structures.

---

[1]MATS, Berkeley, CA, USA [2]Simplex, Astera Institute, Emeryville, CA, USA [3]Beyond Institute for Theoretical Science (BITS), San Francisco, CA, USA. Correspondence to: Paul M. Riechers <pmriechers@gmail.com>, Adam S. Shai <adamimos@gmail.com>.

*Proceedings of the 42nd International Conference on Machine Learning*, Vancouver, Canada. PMLR 267, 2025. Copyright 2025 by the author(s).

## 1. Introduction

Transformers excel at next-token prediction (Vaswani et al., 2017), but their success belies a fundamental tension: optimal prediction requires Bayesian belief updating, a recursive process, while their architecture enforces parallelized, attention-driven computation (Fig. A1). How do transformers resolve this conflict? We show that they develop geometrically structured representations that approximate Bayesian inference under architectural constraints, revealing a precise interplay between theoretical necessity and implementation.

In this work, we combine insights from the theory of optimal prediction with neural network analysis. First, computational mechanics (Shalizi & Crutchfield, 2001a; Marzen & Crutchfield, 2017; Riechers & Crutchfield, 2018b; Pepper, 2024; Shai et al., 2024) dictates *what* an optimal predictor must represent: belief states that encode distributions over futures. Second, mechanistic interpretability reveals *how* transformers approximate these states under architectural constraints, bending Bayesian updates into attention's parallelizable form (Elhage et al., 2021; Nanda et al., 2023).

By combining these frameworks we reveal *why* transformers learn certain intermediate structures. We find that the geometry of a transformer's internal representations is not an accident—it is a mathematical signature of how architectural constraints warp otherwise optimal Bayesian inference. By interpreting learned weights and activations via standard mechanistic interpretability, we uncover an algorithm that is well-captured by the **constrained belief updating** equations. From first principles, we derive the constrained belief geometries, and reverse-engineer the transformer's computational blueprint, predicting attention patterns, value vectors, and residual stream geometries precisely. Thus, beyond verifying that transformers encode belief states, we show how the specific circuits that implement those states necessarily deviate from the unconstrained Bayesian ideal in predictable and theoretically tractable ways.

To concretize these ideas, we focus on transformers trained on data from the Mess3 class of hidden Markov models (HMMs) (Marzen & Crutchfield, 2017), which provides rich and visualizable belief-state geometries and also admits a tractable optimal predictor. Our primary analysis exam-

ines single-layer transformers to isolate how the first attention layer implements constrained belief updating, though we also demonstrate that these principles extend to multi-layer architectures where subsequent layers refine the initial constrained representations. This allows us to rigorously compare the theoretically optimal geometry with the neural-activation geometry that transformers learn. More broadly, we anticipate that the same tension between architecture and optimal inference arises in large language models trained on natural text, and that our methodology would shed light on those more complex cases.

**Key contributions:**

1. **A Unified View of Optimal Prediction and Transformer Computation**: We bridge the model-agnostic theory of Bayesian belief states with the model-specific constraints of attention-based parallel processing. This synthesis explains why transformers trained on next-token prediction discover a distinct "constrained belief updating" geometry—balancing optimal Bayesian inference with the functional form of attention.

2. **Spectral Theory of Constrained Belief Updating**: We develop a theoretical framework that analyzes how eigenvalues of the data-generating transition matrices determine attention heads' behavior. By decomposing belief updates spectrally, we show that multi-head attention naturally implements these scalar updates in orthogonal modes—even handling oscillatory decay of influence—through a sum of specialized head outputs.

3. **Predictive Experiments and Mechanistic Verification**: Our approach yields specific, testable predictions about attention patterns, value vectors, intermediate fractal representations, and final belief-state geometry. We confirm these predictions in trained transformers, demonstrating how the inherently recurrent next-token task is realized by an attention-based, parallelized implementation of Bayesian belief updates.

## 2. Background

### 2.1. Related Work

**Features as directions in activation space**   Modern interpretability research views neural network representations through the lens of linear geometry, analyzing how activation patterns align with specific directions that encode fundamental features (Park et al., 2024). This perspective is particularly useful given superposition (Elhage et al., 2022), where networks encode more features than available neurons using non-orthogonal vectors. Conceptualizing features as linear directions has been instrumental (Cunningham et al., 2023; Bricken et al., 2023; Templeton et al., 2024) in under-

standing what information transformers represent, with geometric relationships between features revealing structured internal representations (Engels et al., 2024). Our work provides a mechanistic explanation for these non-orthogonal geometric structures, providing the theoretical *why* to complement the *what* of feature representations.

**From features to circuits**   While feature directions reveal what information is encoded, understanding how networks process this information is done by identifying computational circuits—subnetworks that implement specific algorithmic operations. Circuits typically combine simpler features into more complex ones as information flows through the network. Examples include circuits that detect syntax (Elhage et al., 2021), implement indirect object identification (Wang et al., 2022), or perform basic arithmetic (Nanda et al., 2023). However, identifying circuits remains largely a manual process, starting from observed behaviors and working backwards to discover relevant components (although active research is developing automated approaches; see Conmy et al. (2023); Marks et al. (2024)).

Our work demonstrates that a principled, top-down theoretical framework, based on constrained belief updating, can guide the search for circuits and provide a deeper understanding of their function within the larger network. We show how specific circuits in the attention mechanism directly implement the computations predicted by our theory.

**Belief state geometry and computational mechanics** Our work draws inspiration from computational mechanics, a framework for studying information processing in dynamical systems (Shalizi & Crutchfield, 2001b; Crutchfield, 2012; Riechers & Crutchfield, 2018b). When applied to sequential data, computational mechanics, in accordance with the POMDP framework (Kaelbling et al., 1998), shows that optimal prediction requires maintaining beliefs about the underlying latent states of the data-generating process (Upper, 1997). These belief states can be visualized as points on a probability simplex, evolving according to Bayesian updating rules, and forming characteristic geometric patterns (Crutchfield, 1994; Marzen & Crutchfield, 2017). Recent work shows that transformers naturally discover and encode these belief state geometries in their activations (Shai et al., 2024). This connection offers a principled way to analyze network representations: rather than reverse-engineering observed behaviors, we can study how architectural constraints shape the network's implementation of theoretically optimal prediction strategies.

This is the approach taken here. We move beyond prior work by proposing and validating a theory of constrained belief updating, demonstrating how specific architectural elements, like the attention mechanism, modify the idealized belief state dynamics. This perspective shifts the focus from

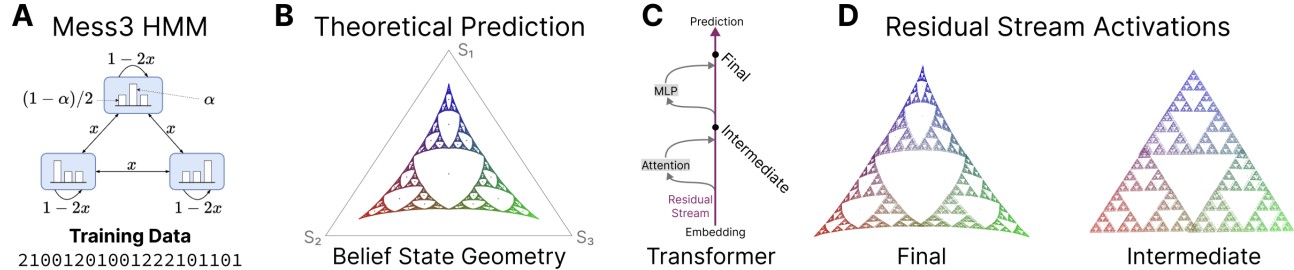

Figure 1: Transformers' internal representations exhibit complex geometric structure matching the belief-state geometry. **(A)** Mess3 HMM, vertices represent hidden states with their emission distributions. **(B)** Ground-truth belief state geometry of Mess3. Each point represents a belief-state probability distribution over hidden states of the HMM, induced via Bayesian updates upon a sequence of observed emissions, with proximity to the vertices of the simplex corresponding to the probabilities of the three hidden states. **(C)** Schematic of a single-layer transformer with Intermediate activations after Attention, and Final activations after the subsequent MLP. **(D)** PCA projections of the model's final residual stream (left), before the unembedding, reveals a geometric representation that closely matches the belief geometry shown in (B), whereas the PCA projection of the intermediate residual stream (right) after attention but before the MLP exhibits an intricate but different structure. In (B) and (D), points are colored according to the ground-truth belief states associated with the sequence of tokens that induces the point, taking the three constituent probabilities over hidden states of the HMM as RGB values.

reverse-engineering learned features to understanding why particular geometric patterns emerge during training as a consequence of the interplay between optimal prediction and architectural constraints. Our work provides a concrete example of how this theoretical framework can be applied to understand the internal mechanisms of transformers.

### 2.2. Optimal Prediction and Belief State Geometry

Shai et al. (2024) showed that transformers minimizing next-token loss internally represent the context-induced probability density over the entire future of possible token sequences:

$$\Pr(Z_{d+1:\infty}|Z_{1:d} = z_{1:d}) \qquad (1)$$

where $Z_{d+1:\infty} = Z_{d+1}, Z_{d+2}, \ldots$ denotes the sequence of random variables for future tokens, $Z_{1:d} = Z_1, \ldots, Z_d$ denotes the sequence of random variables for past tokens, which is realized by a particular sequence of tokens $z_{1:d} \in \mathcal{Z}^d$ known as the context up to position s.

When we conceptualize the training data as being generated by an edge-emitting hidden Markov model (Mealy HMM), we can derive a natural geometric embedding for these conditional probability distributions. HMMs generate training data by emitting tokens when moving among its hidden states $\mathcal{S}$, from one hidden state $S_t$ at time $t$ to the next. The natural geometric embedding is then given by considering how an initial distribution over hidden states $S_0 \sim \boldsymbol{\eta}_\varnothing$, as a point in the vector space $\mathbb{R}^{|\mathcal{S}|}$ (with coordinates given by the probability elements), evolves upon seeing a particular sequence of tokens, $z_{1:d}$. This distribution over the hidden states, which uniquely induces a probability density over all possible futures, is updated

via Bayes rule according to the substochastic transition matrices of the HMM, $\left(T^{(z)}\right)_{z \in \mathcal{Z}}$, with matrix elements $T^{(z)}_{s,s'} = \Pr(Z_{t+1} = z, S_{t+1} = s'|S_t = s)$. In particular, the updated distribution, given context $z_{1:d}$, is

$$\boldsymbol{\eta}_\varnothing \mapsto \vec{r}^{(z_{1:d})}_{\text{full}} = \frac{\boldsymbol{\eta}_\varnothing T^{(z_{1:d})}}{\boldsymbol{\eta}_\varnothing T^{(z_{1:d})}\mathbf{1}} , \qquad (2)$$

where $T^{(z_{1:L})} = T^{(z_1)} \cdots T^{(z_L)}$, and $\mathbf{1}$ is the column vector of all ones. In this paper, we will make the simplifying assumption that the training data is sampled from a stationary stochastic process, in which case the initial distribution over latent states is the stationary distribution $\boldsymbol{\eta}_\varnothing = \boldsymbol{\pi} = \boldsymbol{\pi}T$, where $T = \sum_{z \in \mathcal{Z}} T^{(z)}$ is the row-stochastic transition matrix over hidden states.

Thus, Eq. (2) embeds each token sequence into a probability simplex over the latent states of the HMM—a point in a real-valued vector space. The totality of these points forms a particular geometry, called the belief state geometry, and is universally found in linear form within the activations of various deep neural networks, including RNNs (Pepper, 2024) and transformers (Shai et al., 2024).

This precise framework for anticipating intermediate activations in transformers provides a natural interpretation of the attention mechanism in which it moves information in a belief simplex for the purposes of building up the architecture-independent belief state geometry given in Eq. (2).

## 3. Methodology

**Data Generation.** Our study focuses on the Mess3 parametrized family of hidden Markov models (Marzen

& Crutchfield, 2017), which provide a tractable yet rich setting for studying sequence prediction. As shown in Fig. 1A, these HMMs consist of three hidden states with observable emissions controlled by parameter $\alpha$ and transitions by parameter $x$. Higher values of $\alpha \in [0, 1]$ mean each state more strongly prefers its unique emission symbol, providing clearer information about the generating state. The parameter $x \in (0, \frac{1}{2}]$ controls state persistence—low values create high inertia where states tend to persist, while high values increase transition probabilities between states. For each experimental run, we generate sequences by sampling from an HMM with specific $(\alpha, x)$ values.

**Training Process.** We train single-layer transformers on next-token prediction using gradient descent, with sequences sampled from our parametrized HMMs as training data. The model learns to predict the next token in each sequence by minimizing cross-entropy loss (see App. B for details). Our primary analysis focuses on single-layer architectures to clearly isolate how the attention mechanism implements constrained belief updating, though we also validate that these mechanisms persist as the foundational computation in deeper networks (Figs. A5;A6).

**Analysis of Representations and Computations.** To study how the model processes information, we analyze intermediate and final activations in the residual stream (Fig.1C). We apply principal component analysis (PCA) to these activations across all possible input sequences, finding that the representations are well-captured by a low-dimensional space. In some cases, we slightly rotate the PCA basis to align with theoretically meaningful directions. This dimensionality reduction enables us to visualize how representations evolve through the network—from input embeddings, through the intermediate state after attention, to the final output state after the MLP layer (Fig.1D). To understand how the network manipulates these representations, we analyze the learned weights and attention patterns, examining how attention transforms input embeddings into intermediate representations and how the MLP layer transforms these into the final geometry. At each stage, we compare the learned representations to theoretical predictions derived from optimal Bayesian updates[1].

## 4. Results

### 4.1. Intermediate representations are fractals, but not belief state geometry

Through PCA of the residual stream, we observe two distinct fractal structures in transformers trained on Mess3 HMM data: one after the attention mechanism but before the MLP, and another in the final layer output (Figs. 1, 4). While the

---

[1]Code for analysis of can be found here.

final representations align with the geometry of theoretical belief states, the intermediate fractals exhibit a markedly different structure. The systematic difference between intermediate and final representations raises two key questions: (1) How does attention construct these intermediate fractals and (2) why do they take these particular geometric forms? The following results reveal the algorithmic process behind their construction and provides a theoretical explanation for their previously unexpected structure.

### 4.2. Intermediate representations are built by algorithms in the belief simplex

To determine how the intermediate representation is constructed, we performed mechanistic interpretability on the attention heads. We find that attention performs an algorithm with a direct interpretation in the belief simplex.

At every context position, the residual stream can be thought of as a $d_{\text{model}}$-dimensional skip connection communication channel streaming alongside all layers, carrying all working memory in a transformer (Elhage et al., 2021). Attention and MLP modules read in linear transformations of the residual stream and then add their output to the local residual stream at each layer (Vaswani et al., 2017).

Following (Elhage et al., 2021), we decompose the attention operation into two circuits: (i) the output-value (OV) circuit, which specifies what information is read from each position and how it transforms into a vector that can be broadcast to other positions, and (ii) the query-key (QK) circuit, which computes the similarity of a linearly transformed source and destination to determine how much to update the destination's residual stream with that source's OV contribution.

For an attention head, the residual stream update $\vec{x}_{\text{d}}^{(\text{mid})} = \vec{x}_{\text{d}}^{(\text{pre})} + \vec{c}_{\text{d}} \in \mathbb{R}^{d_{\text{model}}}$ at the *destination* position s is:

$$\vec{c}_{\text{d}} = \sum_{s \leq d} A_{\text{d,s}} \vec{v}_{\text{s}} \qquad (3)$$

Here, $\vec{v}_{\text{s}} = W_O W_V \vec{x}_{\text{s}}^{(\text{pre})}$ represents the OV circuit's contribution from *source* position s, where $W_O$ and $W_V$ are the attention output and value weight matrices respectively, and $\vec{x}_{\text{s}}^{(\text{pre})}$ is the incoming residual stream vector at position s. Attention $A_{\text{d,s}}$ is determined by the QK circuit through query–key inner product and the causally masked softmax operations:

$$A_{\text{d,s}} = \delta_{s \leq d} \frac{e^{\vec{q}_{\text{d}} \cdot \vec{k}_{\text{s}} / \sqrt{d_{\text{h}}}}}{\sum_{s'=1}^{\text{d}} e^{\vec{q}_{\text{d}} \cdot \vec{k}_{s'} / \sqrt{d_{\text{h}}}}} \ , \qquad (4)$$

where $\vec{q}_{\text{d}} = W_Q \vec{x}_{\text{d}}^{(\text{pre})}$ is the query vector from position d, $\vec{k}_{\text{s}} = W_K \vec{x}_{\text{s}}^{(\text{pre})}$ is the key vector from position s, $d_{\text{h}}$ is the head dimension, and $W_Q$ and $W_K$ are each $d_{\text{h}} \times d_{\text{model}}$ weight matrices. Recall that attention is non-negative $0 \leq A_{\text{d,s}} \leq 1$

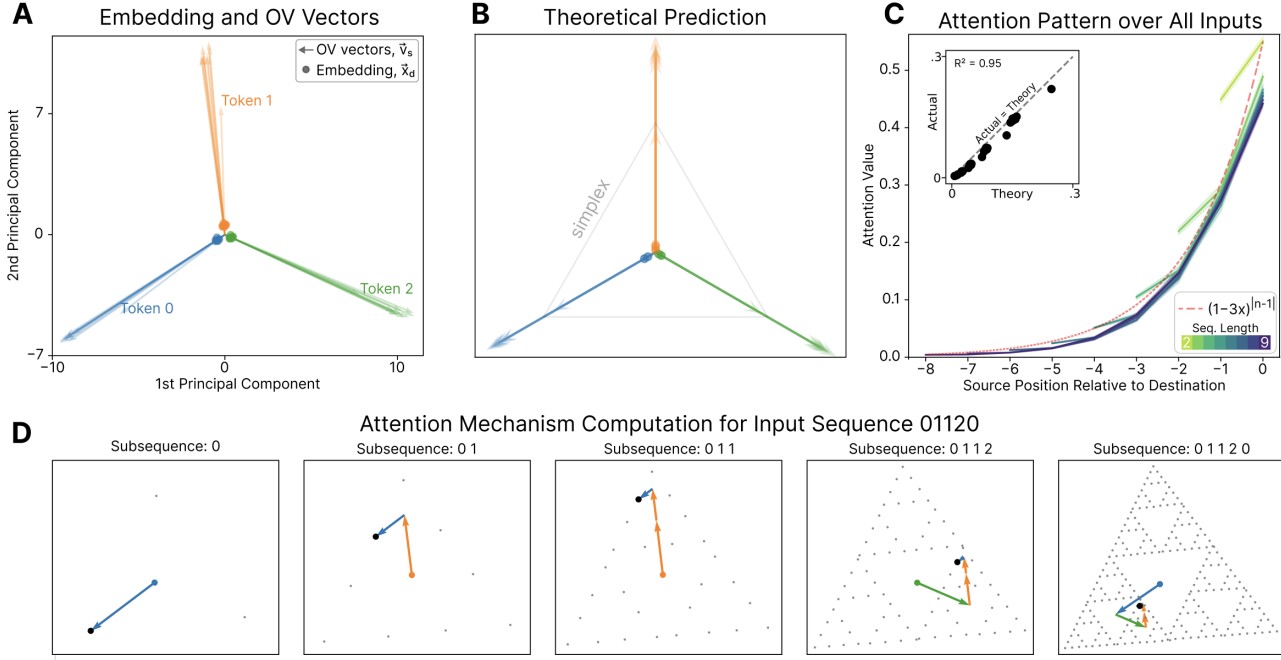

Figure 2: **Intermediate Representation Construction by Attention.** A transformer trained on Mess3 with $x = 0.15$ and $\alpha = 0.6$ exhibits intermediate representations constructed through a specific attention mechanism. **(A)** The OV vectors (arrows) form three distinct clusters, each corresponding to a token and positioned at the vertices of a triangle, while token embeddings (circles) are clustered near the origin. **(B)** Our theoretical predictions for the OV vectors (shown for all (position, token) pairs) and embeddings (for positions $> 2$) align closely to those found in the trained transformer. **(C)** Attention patterns are primarily determined by the positional distance between the destination and source tokens, following an exponential decay described by $(1 - 3x)^{|n-1|}$. They are largely independent of specific token sequences. **(C, inset)** The theoretical (Eq. (13)) and actual values in the attention pattern align closely. **(D)** Construction of intermediate representations for five input subsequences of increasing length (from the example sequence 01120, shown left to right). The attention mechanism builds the fractal by taking linear combinations of the three $\vec{v}_s$ vectors. The colored vectors illustrate the components of the sum for each example subsequence, while the gray dots represent all possible vector sums for all sequences at that position.

and, for each destination position d, the attention to all sources sums to one: $\sum_{s \leq d} A_{d,s} = 1$. Eq. (3) shows how each attention head computes its update by weighting the transformed values ($\vec{v}_s$) from all previous positions according to their relevance ($A_{d,s}$) to the current position.

Our analysis yields several key insights into how the attention mechanism constructs the intermediate representations. First, we find that projecting token embeddings (the inputs into the attention head) onto PCA space reveals three clusters that lie close to the origin, as shown in Fig.2A. Meanwhile, the OV projections form update vectors $\vec{v}_s$ that cluster in three directions pointing toward the vertices of a triangle, naturally interpreted as the vertices of the belief simplex in Fig. 2. The model combines these directions through weights $A_{d,s}$ determined by the QK circuit as described by Eq. (3). For Mess3, these attention weights are invariant to token identity and decay exponentially with distance from the current position, controlling how past information is

integrated. As the attention weight decays with distance, the impact of past tokens on the current belief state diminishes over time. Through this process of weighted vector addition within the belief simplex, the attention mechanism constructs the intermediate representations, resulting in the observed fractal structure shown in Fig. 2D. Incredibly, **the computation the attention head performs is completely interpretable as a dynamic process in the belief simplex**.

### 4.3. Relating Intermediate Representations to Belief Updating Equations

The interpretation of attention as operating in the belief simplex suggests a connection to the theory of belief updating. Since the OV circuit is only able to access information from the source token that is attended to, we can write a constrained belief updating equation that sums contributions from the value of the token $n = d - s$ places back for each value of $n$, assuming the initial belief is the stationary distri-

bution of the HMM, $\boldsymbol{\pi}$. This gives the following equation for the constrained belief at position d in the sequence:

$$\vec{r}_1^{(z_{1:d})} = \boldsymbol{\pi} + \sum_{s=1}^{d} \left( \boldsymbol{\pi} T^{|z_s} T^{d-s} - \boldsymbol{\pi} \right) \qquad (5)$$

where $T$ is the HMM's hidden state transition matrix (marginalizing out the emissions), and $T^{|z}$ is the HMM transition matrix conditioned on seeing token $z$ (see App. A for details). Eq. (5), interpreted as a context-induced point in a vector space, is the natural geometric embedding of

$$\Pr(S_d) + \sum_{s=1}^{d} \left[ \Pr(S_d | Z_s = z_s) - \Pr(S_d) \right] . \qquad (6)$$

This equation describes the best possible embedding if you haven't seen any context, $\Pr(S_d) = \boldsymbol{\pi}$, followed by independent corrections to that prediction from the token at each preceding context position, $\Pr(S_d | Z_s = z_s) - \Pr(S_d) = \boldsymbol{\pi} T^{|z_s} T^{d-s} - \boldsymbol{\pi}$. Notably, since Eq. (6) is a distribution over latent states $S_d$ rather than merely the next token $Z_{d+1}$, this constrained-update equation naturally implemented by attention implies a probability density over all extended futures $Z_{d+1:\infty}$ rather than just the next timestep.

It is useful to take a step back and get a handle on the intuition for Eq. (5) and Eq. (6). Bayesian inference (Eq. (2)) requires multiplying token-specific transition matrices, a fundamentally recursive process where updates depend on the full history integrated up to the previous step. In contrast, an attention head computes its output at position d via a parallel, feedforward weighted sum (Eq. 3) of value vectors, $v_s$ from source positions $s \leq d$. Crucially, each $v_s$ contains only local information from position s. It cannot directly access or depend on the specific tokens between s and d due to the parallel nature of the value computation and attention weighting. Therefore, the most information that token $z_s$ can independently contribute to the belief at position d within this single-layer constraint is the correction derived from knowing $z_s$ occurred $d - s$ steps prior, assuming a default starting belief $\pi$ and no knowledge of intervening tokens. These independent displacements from the stationary distribution over latent states is the difference of probability distributions: $Pr(S_d | Z_s) - Pr(S_d)$. Linear algebraically, this contribution is precisely $\pi T^{|z_s} T^{d-s} - \pi$. Summing these over all past sources naturally yields the constrained belief update form in Eq. (5) (see Fig. A1 for a conceptual diagram). It represents the best possible parallel approximation achievable by a single attention layer given its architectural limitations.

Eq. (5)'s constrained belief geometry closely matches the intermediate structure observed in the central range of $\alpha \in [0.2, 0.6]$, as shown in the left two columns in Fig. 4. As $\alpha$ moves further from this range, we observe gradually

increasing deviations between predicted and actual representations, though the overall structure remains similar. A complete characterization of how these deviations scale with $\alpha$ remains for future work.

Fig. A3 demonstrates that transformers reliably discover these theoretical geometries during training, with MSE to the constrained belief theory decreasing rapidly for post-attention activations while MSE to the full belief geometry simultaneously decreases for post-MLP activations. The final converged representations show strong quantitative agreement with our theoretical predictions (Fig. A4), with MSE values orders of magnitude lower than random initialization for both the constrained and full belief geometries.

### 4.4. Attention Implements a Spectral Algorithm to Build the Constrained Beliefs

Eq. (5) shows how the attention pattern in our model must relate to powers of the Markov transition matrix of the underlying hidden states, $T^n$, where $n$ is the relative token distance.

To understand how this works, we turn to spectral analysis. The main goal is to understand how information or influence from a past token (at source position s) propagates to affect the belief state at the current destination position d. This influence mathematically depends on the sequence of hidden state transitions between s and d, captured by powers of the HMM transition matrix, $T^{d-s}$.

Spectral decomposition (using eigenvalues $\lambda$ and associated projectors $T_\lambda$) is a standard mathematical tool to analyze matrix powers because it simplifies $T^n$ into a sum $\sum_\lambda \lambda^n T_\lambda$. The eigenvalues ($\lambda$) are crucial because they tell us the rate at which the influence of past information decays (if $|\lambda| < 1$) or even oscillates (if $\lambda$ is negative or complex) as the distance $n = d - s$ increases.

Our key finding is that the learned attention weights ($A_{d,s}$ in Eq. 3) directly implement this propagation effect, effectively learning to approximate the $\lambda^{d-s}$ decay predicted by the theory. This explains why attention patterns often show exponential decay, and why multiple heads are needed to capture oscillatory patterns arising from negative eigenvalues. Furthermore, this spectral perspective allows us to make precise, verifiable predictions about the learned OV vectors and token embeddings, directly connecting the dynamics of the data (via $T$'s eigenvalues) to the specific parameters learned by the transformer.

When $T$ is diagonalizable with a set of eigenvalues $\Lambda_T$, it then has a simple spectral decomposition such that we can

rewrite Eq. (5) as

$$\vec{r}_1^{(z_{1:d})} = \boldsymbol{\pi} + \sum_{s=1}^{d} \sum_{\lambda \in \Lambda_T \setminus \{1\}} \lambda^{d-s} \boldsymbol{\pi} T^{|z_s} T_\lambda \qquad (7)$$

where $T_\lambda$ is the spectral projection operator associated with eigenvalue $\lambda$ (Riechers & Crutchfield, 2018a). In this diagonalizable case, $T_\lambda = \sum_{k=1}^{a_\lambda} |\lambda_k\rangle\langle\lambda_k|$, where $a_\lambda$ is the algebraic multiplicity of the eigenvalue $\lambda$, with right eigenstates satisfying $T |\lambda_k\rangle = \lambda |\lambda_k\rangle$, left eigenstates satisfying $\langle\lambda_k| T = \lambda \langle\lambda_k|$, all satisfying the orthonormality condition $\langle\lambda_j|\lambda_k\rangle = \delta_{j,k}$. Notably in Eq. (7), all dependence on inter-token distance now lies solely in the exponentiation of the eigenvalues, which all live on or within the unit circle in the complex plane for a stochastic transition matrix like $T$.

For the Mess3 process, the stochastic matrix $T$ has eigenvalues $\Lambda_T = \{1, \zeta\}$, where $\zeta = 1 - 3x$ is a degenerate eigenvalue with multiplicity $a_\zeta = 2$. We observed that the attention weight $n$ tokens back is approximately $\zeta^n = (1-3x)^n$, which suggests a strong connection between the theoretically motivated Eq. (7) and the architectural-implementation Eq. (3). Encouraged by this correspondence and further evidence of similarity, we make the ansatz that *the role of attention in the first layer is to implement the constrained belief update of Eq. (6) via Eq. (7)'s spectral mechanism* [2]. Taking this ansatz seriously allows us to precisely anticipate the analytic form of the learned attention pattern.

To derive the analytic form of the attention pattern, we assume that there is a linear map $f : \mathbb{R}^{d_{\text{model}}} \to \mathbb{R}^{|\mathcal{S}|-1}$ from the residual stream to the hyperplane containing the probability simplex over the hidden states of a minimal generative model of the data (the 2-simplex in this case). Let $\Pi_{\boldsymbol{\Delta}} = I - T_1 = I - \mathbf{1}\boldsymbol{\pi}$ be the projection from $\mathbb{R}^{|\mathcal{S}|}$ to the hyperplane $\mathbb{R}^{|\mathcal{S}|-1}$ containing the simplex. Our full ansatz is thus $f(\vec{x}_d^{(\text{mid})}) = \vec{r}_1^{(z_{1:d})} \Pi_{\boldsymbol{\Delta}}$ or, more explicitly:

$$f(\vec{x}_d^{(\text{mid})}) = \sum_{s=1}^{d} \sum_{\lambda \in \Lambda_T \setminus \{1\}} \lambda^{d-s} \boldsymbol{\pi} T^{|z_s} T_\lambda \qquad (8)$$

$$= f(\vec{x}_d^{(\text{pre})}) + \sum_{s \leq d} A_{d,s} f(\vec{v}_s) . \qquad (9)$$

From this, we group source-specific terms to infer that

$$f(\vec{x}_d^{(\text{pre})}) + A_{d,d} f(\vec{v}_d) = \boldsymbol{\pi} T^{|z_d} - \boldsymbol{\pi} \qquad (10)$$

and

$$A_{d,s} f(\vec{v}_s) = \sum_{\lambda \in \Lambda_T \setminus \{1\}} \lambda^{d-s} \boldsymbol{\pi} T^{|z_s} T_\lambda \quad \text{for d} > \text{s} . \qquad (11)$$

From Eq. (11), we notice that $f(\vec{v}_s)$ is in the linear span of the non-stationary left eigenstates of $T$. I.e., $f(\vec{v}_s) \in$

span$\big(\{\langle\lambda| : \lambda \langle\lambda| = T \langle\lambda| \text{ and } \lambda \neq 1\}\big)$ and, in particular, $f(\vec{v}_s) \cdot |1\rangle = 0$ such that *adding any of the OV vectors to any stochastic vector (whose elements by definition add to one) keeps you in the hyperplane of the probability simplex.*

For the Mess3 family of processes, $T$ has a single eigenvalue $\zeta = 1 - 3x$ with multiplicity $a_\zeta = 2$ besides its eigenvalue of 1. Accordingly, Eq. (11) simplifies to

$$A_{d,s} f(\vec{v}_s) = \zeta^{d-s} \boldsymbol{\pi} T^{|z_s} T_\zeta \qquad \text{for d} > \text{s} , \qquad (12)$$

which forces $f(\vec{v}_s) = c\boldsymbol{\pi} T^{|z_s} T_\zeta$ for some $c \in \mathbb{R}$ independent of d, from which we obtain

$$A_{d+m,s} = \zeta^m A_{d,s} \qquad \text{for d} > \text{s} . \qquad (13)$$

So, for example, $A_{2,1}$ implies $A_{d,1}$ for all destinations d $\geq 2$; and $A_{3,2}$ implies $A_{d,2}$ for all destinations d $\geq 3$.

For Mess3, $T_\zeta = I - |1\rangle\langle 1| = I - \mathbf{1}\boldsymbol{\pi}$, since all projection operators must sum to the identity. Combining this insight with Eq. (12) tells us about the OV-vector for all positions:

$$f(\vec{v}_m) = \frac{\zeta}{A_{m+1,m}} \big(\boldsymbol{\pi} T^{|z_m} - \boldsymbol{\pi}\big) . \qquad (14)$$

Notably, Eq. (14) tells us that all OV-vectors associated with the same token must be parallel—$f(\vec{v}_s) \propto f(\vec{v}_{s'})$ if $z_s = z_{s'}$—which is consistent with what we observe in our experiments (Fig. 2A). Moreover, the magnitude of the $m^{\text{th}}$ OV-vector is inversely proportional to the attention element $A_{m+1,m}$, which is again consistent with our experiments (Fig. 2AB). In our experiments, we find $A_{2,1}$ to be significantly larger than all the other $A_{m+1,m}$ elements, while the latter all cluster together; the magnitude of $\vec{v}_1$ is correspondingly smaller than all of the other strongly clustered $\vec{v}_m$ magnitudes.

Combining Eqs. (10) and (14) constrains the embedding

$$f(\vec{x}_m^{(\text{pre})}) = \left(1 - \frac{\zeta A_{m,m}}{A_{m+1,m}}\right)\big(\boldsymbol{\pi} T^{|z_m} - \boldsymbol{\pi}\big) \qquad (15)$$

to be parallel to the OV-vectors, as we indeed observe.

Eqs. (13), (14) and (15) make *strong predictions about the form of the attention pattern and how it relates to OV-vectors and token embeddings*, which must be true if the first layer of attention is indeed implementing the constrained belief updates over latent states of a generative model of the training data. These relationships are all borne out in our experiments (Fig. 2ABC), except for some scalar discrepancy in the first two embedding vectors (see Appendix D for quantification), which is a strong validation of the predictive power of our framework.

### 4.4.1. NEGATIVE EIGENVALUES REQUIRE MORE ATTENTION HEADS

For the transition matrix $T$ to be row stochastic (a requirement for a valid HMM), $x$ must be in the range $[0, 0.5]$.

---

[2]The details of this correspondence break down if there are many attention heads in the first layer.

Interestingly, when $\zeta < 0$ (which occurs when $x > 1/3$), the predicted pattern oscillates and cannot be captured by a single attention head, since attention pattern entries must be non-negative. In these cases, we observe that a single-head transformer captures an incomplete representation of the belief state geometry, and the transformer performs correspondingly worse (App. F). However, upon adding a second attention head, the model converges to the solution predicted by the belief updating equation, even in the presence of oscillatory dynamics, as shown in Fig. 3.

The anticipated need for a second attention head when the data-generating transition matrix has a negative eigenvalue further demonstrates how our analysis provides a handle to relate the architectural constraints of the attention mechanism to the structure of the training data. In fact, our framework provides more specific predictions for the attention pattern and its relation to embedding and OV-vectors in this case too.

With two attention heads, the update to the residual stream at the destination position s becomes

$$\vec{c}_\mathrm{d} = \sum_{s=1}^{d} \sum_{h=1}^{2} A_{\mathrm{d},s}^{(h)} \vec{v}_s^{(h)} \;, \tag{16}$$

where each head now has its own QK and OV matrices. With the negative eigenvalue $\zeta < 0$ and two attention heads, we can relate the constrained belief update to the details of attention and embedding via

$$f(\vec{x}_\mathrm{d}^{(\mathrm{mid})}) = \sum_{s=1}^{d} (-1)^{\mathrm{d-s}} (-\zeta)^{\mathrm{d-s}} \boldsymbol{\pi} T^{|z_\mathrm{s}} T_\zeta \tag{17}$$

$$= f(\vec{x}_\mathrm{d}^{(\mathrm{pre})}) + \sum_{s \le d} \left[ A_{\mathrm{d},s}^{(1)} f(\vec{v}_s^{(1)}) + A_{\mathrm{d},s}^{(2)} f(\vec{v}_s^{(2)}) \right] \;.$$

This is naturally accommodated by

$$A_{\mathrm{d},s}^{(1)} f(\vec{v}_s^{(1)}) = +\delta_{+1,(-1)^{\mathrm{d-s}}} |\zeta|^{\mathrm{d-s}} \boldsymbol{\pi} T^{|z_\mathrm{s}} T_\zeta \quad \text{and} \tag{18}$$

$$A_{\mathrm{d},s}^{(2)} f(\vec{v}_s^{(2)}) = -\delta_{-1,(-1)^{\mathrm{d-s}}} |\zeta|^{\mathrm{d-s}} \boldsymbol{\pi} T^{|z_\mathrm{s}} T_\zeta \tag{19}$$

for $\mathrm{d} > \mathrm{s}$, which implies that the OV-vectors point in opposite directions, $\widehat{f(\vec{v}_s^{(1)})} = -\widehat{f(\vec{v}_s^{(2)})}$, with $f(\vec{v}_s^{(h)}) \propto (\boldsymbol{\pi} T^{|z_\mathrm{s}} - \boldsymbol{\pi})$ and

$$A_{\mathrm{d}+2m,s}^{(h)} = \zeta^{2m} A_{\mathrm{d},s}^{(h)} \qquad \text{for d} > \mathrm{s} \;, \tag{20}$$

consistent with our experiments as shown in Fig. 3. We note that the magnitudes of OV vectors are tied to attention magnitudes via $c\zeta^{\mathrm{d-s}} = A_{\mathrm{d},s}^{(1)} |f(\vec{v}_s^{(1)})| - A_{\mathrm{d},s}^{(2)} |f(\vec{v}_s^{(2)})|$, with $c = |\boldsymbol{\pi} T^{|z_\mathrm{s}} - \boldsymbol{\pi}| \in \mathbb{R}$, which is also observed in Fig. 3.

### 4.5. Post-MLP geometries

While the intermediate geometry is well characterized by our constrained belief equations, the MLP transformation is

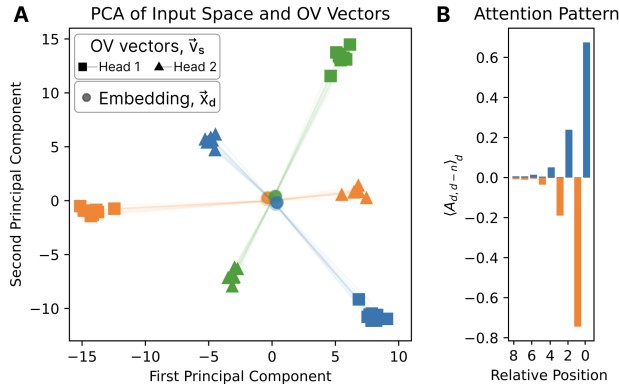

**A**    PCA of Input Space and OV Vectors    **B**    Attention Patterns

Figure 3: Attention heads combine to capture oscillatory dynamics in belief updating. (A) In the token embedding space, the model uses each attention head to embed tokens on opposite poles of the simplex. (B) The attention patterns of the two heads (shown here averaged over all sequences) act as positive and negative components. When combined, they produce the oscillatory pattern predicted by the exponentiated eigenvalue $\zeta^n = (1 - 3x)^n = (-1)^n (3x - 1)^n$.

more complex. Through purely local computations at each position, the MLP learns a continuous nonlinear warping that transforms the intermediate fractal structure into the final belief geometry.

Fig. 4 shows the close match between theoretical predictions and observed representations across different Mess3 parameter settings, confirming our theoretical understanding. The transformation involves stretching and compressing different regions of the space while maintaining topological structure, though a full characterization of its mathematical properties remains for future work.

To verify that our single-layer analysis captures the fundamental computation even in deeper networks, we analyzed 4-layer transformers trained on the same Mess3 data. We find that the first attention layer consistently implements the constrained belief update mechanism, Eq. (5), with subsequent layers progressively transforming the representation toward the full Bayesian belief geometry, Eq. (2). Quantitative MSE analysis (Fig. A6) shows that first-layer attention outputs align well with constrained beliefs while later layers systematically improve alignment with full Bayesian beliefs. This progression is shown in Fig. A5, confirming that our theoretical framework for the first layer remains the foundational computation in multi-layer networks.

## 5. Discussion and Conclusion

We have shown how combining **computational mechanics** with **mechanistic interpretability** yields a principled understanding of why transformers trained on Mess3 HMM

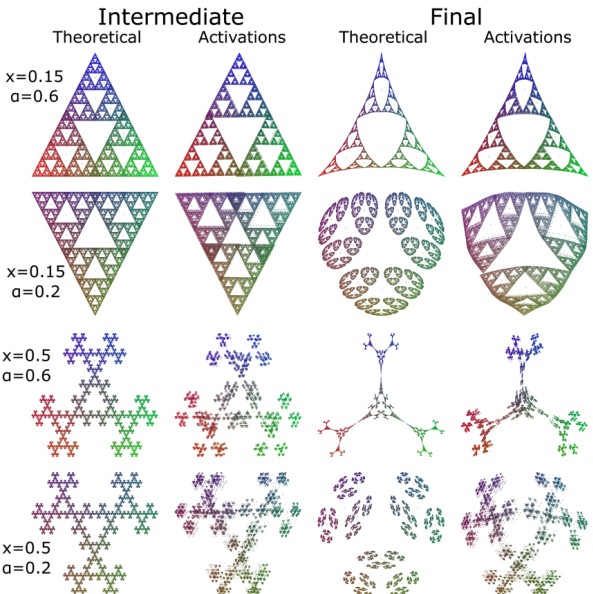

Figure 4: Comparison of model representations and theoretical predictions for different Mess3 hyperparameters in each row. Each subfigure shows four columns: (i) Intermediate representation from Eq. (5). (ii) PCA projection of the model activations in the intermediate layer. (iii) Ground truth belief state geometry from Eq. (2). (iv) PCA projection of the final activations after the MLP.

data learn intermediate fractal-like structures, and how these structures systematically transition into final belief-state representations. By focusing primarily on single-layer transformers, we isolated the fundamental computation performed by the attention mechanism, while also demonstrating that these principles extend to the first layer of deeper networks. This approach provides a top-down theoretical explanation grounded in the tension between optimal Bayesian belief updates and the parallel, attention-based constraints of transformer computation, developing geometric observations of activation space into mechanistic understanding of the underlying computational principles.

**Implications for interpretability.** Our work demonstrates an alternative to bottom-up architectural analysis. Knowing the structure of optimal predictors allows us to predict and verify the *specific* intermediate computations that are implemented by the attention mechanism. Our analysis reveals the computational role of specific directions in activation space—showing how the geometry of belief updates shapes the learned representations. Additionally, by focusing on a small, tractable HMM, we see how specific properties of its transition matrix lead to oscillatory patterns that require specialized multi-head solutions due to the non-negativity constraints of attention mechanisms. Rather than relying

on general observations that attention heads specialize, our analysis reveals precisely *why* and *how* multiple heads must coordinate: the non-negativity constraints of attention, combined with oscillatory patterns in optimal belief updates, necessitate specific decompositions across heads, providing concrete mechanistic understanding of their functional roles. This demonstrates how combining theoretical understanding with architectural constraints can yield precise, verifiable interpretations of neural network components.

**Limitations and future work.** Our experiments used small transformers (1 and 4 layers), with analysis focused on the first attention layer. For training data we used the specialized Mess3 family of HMMs with full support over the space of all possible sequences of tokens. We discovered how transformers implement belief updates when attention patterns depend primarily on positional distances, while token-specific information is handled through value vectors. While we validated that the first layer of multi-layer transformers implements the same constrained belief updating mechanism (Figs. A5;A6), with subsequent layers refining toward full Bayesian beliefs, our techniques must be adapted to both larger transformer architectures and data-generating processes that capture the complexities of real-world data. While this setting offers clear insights, it does not capture many aspects of natural language. Future work could apply these techniques to processes that better reflect properties of natural language—hierarchical, with sparse support over sequences. Moreover, the interplay between multi-head attention and deeper layer stacks likely exhibits additional nuances that our primary single-layer analyses only begins to uncover. Finally, while we showed that the final MLP layer refines partial updates to approximate full Bayes, the deeper question of why gradient descent converges on these circuits remains ripe for further investigation.

**Conclusion.** By combining computational mechanics with mechanistic interpretability, we have shown how transformers implement inherently recursive Bayesian updates through parallel computations via the attention mechanism, and how these intermediate representations are refined into the final form. This reconciles model-agnostic theories of next-token prediction with the reality of architecture-specific constraints. We hope our results not only advance interpretability for HMM-like toy tasks but also inspire deeper theoretical insights into how large-scale transformers produce—and exploit—belief-like structures in real-world applications.

## Acknowledgments

The authors are grateful for the community and financial support from MATS, PIBBSS, FAR Labs, BITS, and Astera Institute, and MP's further financial support from Open

Philanthropy during the MATS extension program, which made this project possible.

**Author Contributions**

MP discovered the attention-based constrained belief updating algorithm in the simplex, and performed the bulk of the experiments with mentorship from ASS. PMR developed the mathematical theory together with MP and ASS. ASS supervised the project, and DF provided project management. MP, PMR, and ASS wrote the manuscript, with helpful guidance from DF. MP, PMR, and ASS performed analysis, and established the correspondence between transformer behavior and theoretical predictions.

## Impact Statement

This paper presents work whose goal is to advance the interpretability of Machine Learning. There are many potential societal consequences of our work, none which we feel must be specifically highlighted here.

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

# A. Mathematical Details of HMMs and Belief State Geometry

In this work we created training data from a class of Hidden Markov Models (HMMs) called Mess3. The HMMs have three hidden states $\mathcal{S} = \{1, 2, 3\}$ and emit from a vocabulary of three tokens $\mathcal{Z} = \{0, 1, 2\}$.

The HMMs in this class are parameterized by $\alpha$ and $x$, with dependent quantities $\beta = (1 - \alpha)/2$ and $y = 1 - 2x$.

The labeled transition matrices define the probability of moving to state $j$ (indexing columns) and emitting the token on the label, $z$, conditioned on being in state $i$ (indexing rows), $P(s_j, z|s_i)$ and are:

$$T^{(0)} = \begin{bmatrix} \alpha y & \beta x & \beta x \\ \alpha x & \beta y & \beta x \\ \alpha x & \beta x & \beta y \end{bmatrix} \tag{21}$$

$$T^{(1)} = \begin{bmatrix} \beta y & \alpha x & \beta x \\ \beta x & \alpha y & \beta x \\ \beta x & \alpha x & \beta y \end{bmatrix} \tag{22}$$

$$T^{(2)} = \begin{bmatrix} \beta y & \beta x & \alpha x \\ \beta x & \beta y & \alpha x \\ \beta x & \beta x & \alpha y \end{bmatrix} \tag{23}$$

Note that even though the dynamics amongst the emissions are infinite-Markov order, the dynamics amongst the hidden states are Markov, with a transition matrix given by marginalizing out the token emissions: $T = \sum_{z \in \mathcal{Z}} T^{(z)}$.

Since Mess3 has non-zero row sums for each labeled transition matrix, we can also define a conditional transition matrix, $T^{|z}$, with elements $T_{i,j}^{|z} = P(s_j|z, s_i)$, which is given by normalizing each labeled transition matrix such that every row sums to 1.

## A.1. Full belief updates

An important part of the work presented here is about how an optimal observer of token emissions from the HMM would update their beliefs over which of the hidden states the HMM is in, given a token sequence. If the observer is in a belief state given by a probability distribution $\boldsymbol{\eta}$ (a row vector) over the hidden states of the data-generating process, then the update rule for the new belief state $\boldsymbol{\eta}'$ given that the observer sees a new token $z$ is:

$$\boldsymbol{\eta}' = \frac{\boldsymbol{\eta} T^{(z)}}{\boldsymbol{\eta} T^{(z)} \mathbf{1}} \tag{24}$$

where $\mathbf{1}$ is a column vector of ones of appropriate dimension, with the denominator ensuring proper normalization of the updated belief state. In general, starting from the initial belief state $\boldsymbol{\eta}_\varnothing$, we can find the belief state after observing a sequence of tokens $z_0, z_1, \ldots, z_N$:

$$\vec{r}_{\text{full}}^{(z_{1:d})} = \frac{\boldsymbol{\eta}_\varnothing T^{(z_0)} T^{(z_1)} \cdots T^{(z_N)}}{\boldsymbol{\eta}_\varnothing T^{(z_0)} T^{(z_1)} \cdots T^{(z_N)} \mathbf{1}} . \tag{25}$$

For stationary processes, the optimal initial belief state is given by the stationary distribution $\boldsymbol{\eta}_\varnothing = \boldsymbol{\pi}$ over hidden states of the HMM (the left-eigenvector of the transition matrix $T = \sum_z T^{(z)}$ associated with the eigenvalue of 1).

The beliefs have a geometry associated with them, called the belief-state geometry. The belief-state geometry is given by plotting the belief distribution over the HMM's hidden states induced from each possible sequence of tokens as a point in the probability simplex over these hidden states.

## A.2. Constrained belief updates

Incorporating past contributions to belief updates in parallel, as the attention mechanism suggests, we instead obtain

$$\vec{r}_1^{(z_{1:d})} = \boldsymbol{\pi} + \sum_{n=0}^{d-1} \left( \frac{\boldsymbol{\pi} T^{(z_{d-n})} T^n}{\boldsymbol{\pi} T^{(z_{d-n})} \mathbf{1}} - \boldsymbol{\pi} \right) \tag{26}$$

For processes like Mess3 that have non-zero row sums for each labeled transition matrix, this can be written more simply as:

$$\bar{r}_1^{(z_{1:\mathrm{d}})} = \boldsymbol{\pi} + \sum_{n=0}^{\mathrm{d}-1} \left( \boldsymbol{\pi} T^{|z_{\mathrm{d}-n}} T^n - \boldsymbol{\pi} \right) , \tag{27}$$

which is the form that appears in the main text. For other processes that don't satisfy this condition, slight modifications of the equations in the main text follow straightforwardly from Eq. (26).

## B. Model architecture and training procedure

We employ a standard single-layer transformer model with learned positional embeddings. The model architecture follows the conventional transformer design, with $d_{\mathrm{model}} = 64$ and $d_{\mathrm{ff}} = 256$. Depending on the Mess3 parameters, we use either a single-head or a double-head attention mechanism. We conduct a systematic sweep over the HMM parameters $\alpha$ and $x$, training a separate model for each pair. Models are trained on next-token prediction using cross-entropy loss, with batch size 128. We use Adam optimizer (Kingma & Ba, 2017) with a $10^{-4}$ learning rate and no weight decay. Each model is trained for approximately 15 million tokens.

We generate all possible input sequences up to length 10, recording hidden activations from the transformer's residual stream. These activations are organized into a dataset capturing the model's response to all input patterns.

Input sequences consist of three symbols, embedded with positional information, without a beginning-of-sequence (BOS) token.

## C. Recurrent vs. Parallel Belief Updating

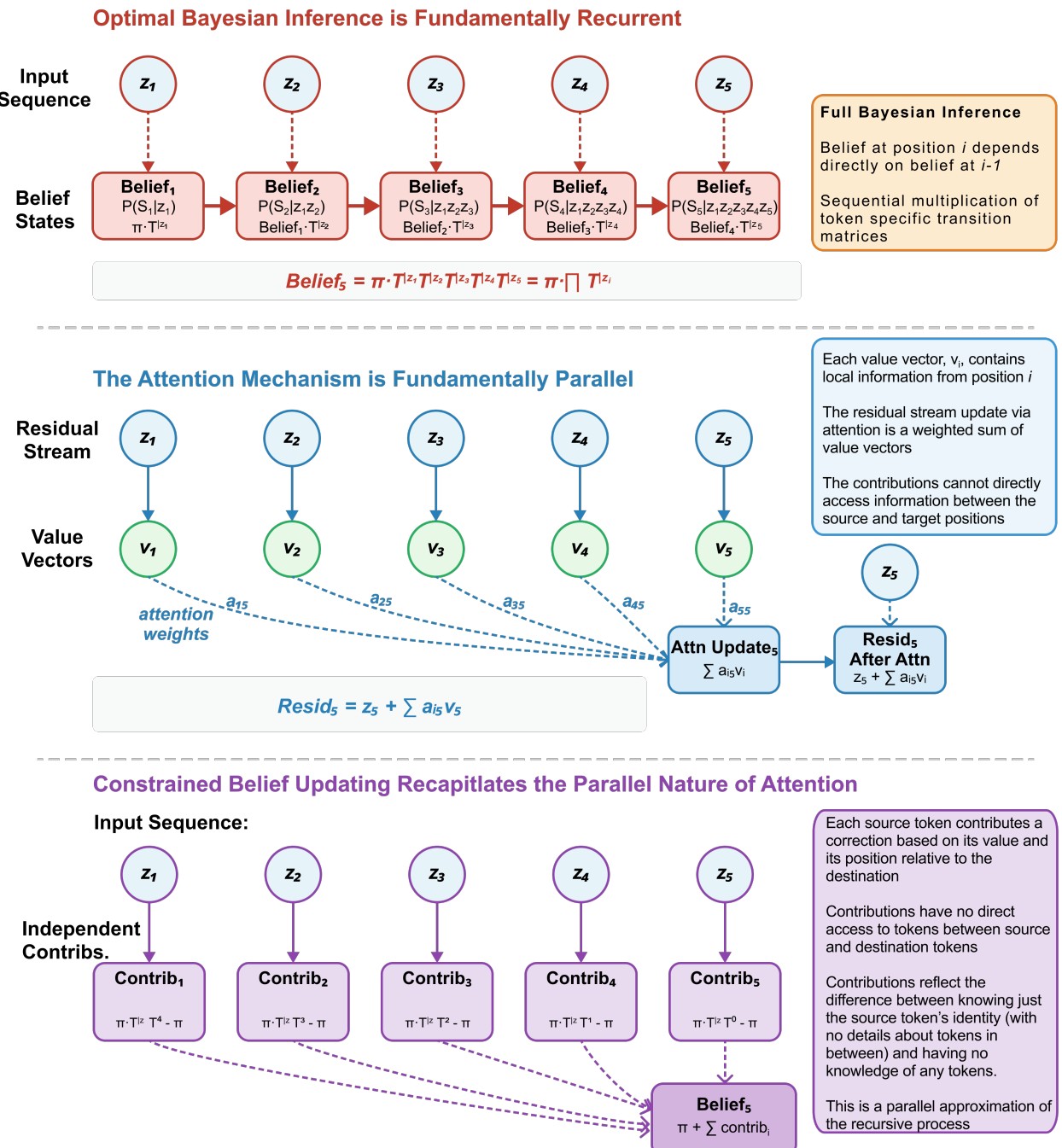

Figure A1: Diagrammatic visualization comparing the recurrent nature of Optimal Bayesian Inference (top), to the parallel attention mechanism (middle), and the parallel Constrained Belief Updating presented in this paper.

## D. Quantification of Theoretical Predictions

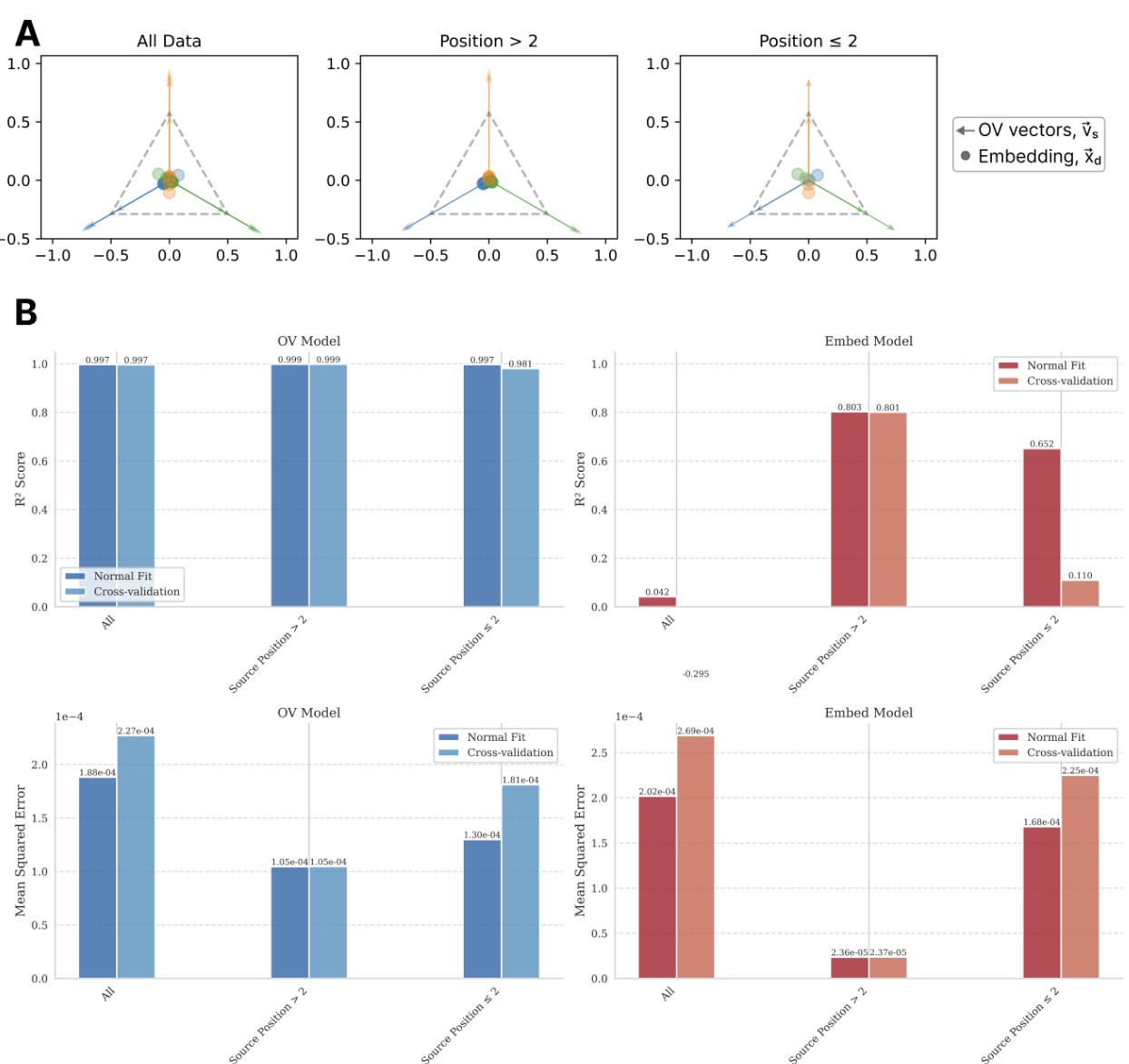

Figure A2: Embeddings for the first two positions are correctly predicted to be parallel to the OV vectors, as with all of the embeddings; however the sign of the predicted embedding for these first two positions deviates from the observed embedding. We do not yet understand the reason for this discrepancy, but still find it remarkable that the bulk of the high-dimensional computation carried out by attention—attention pattern, OV vectors, and all embeddings beyond the first two positions—can be very precisely understood by a sequence of operations in the two-dimensional simplex.

## E. Multi-layer experiments

## F. Minimal architectural requirements

To verify our theoretical understanding of the transformer's computational requirements, we conduct a systematic evaluation across different architectural configurations. Fig. A7 shows that the model achieves good performance with minimal architecture: a single layer with two attention heads is sufficient to achieve low KL divergence across different Mess3

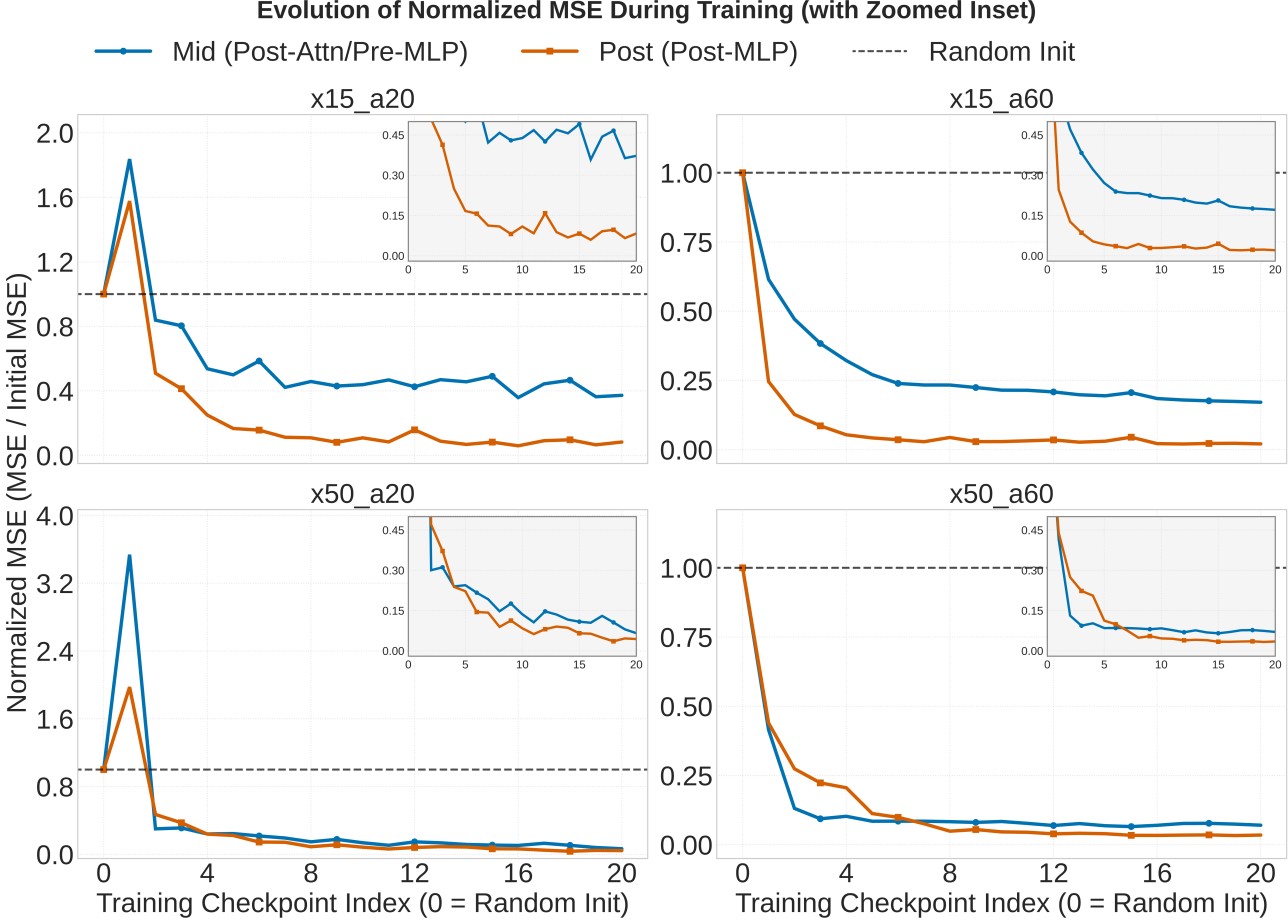

Figure A3: **Evolution of Normalized Mean Squared Error (MSE) During Training Across Experimental Conditions.**
Each subplot shows the MSE for one experimental condition (labeled by title), normalized by the initial MSE value from the randomly initialized model (represented at Checkpoint Index 0). The Y-axis represents MSE relative to this random baseline (Lower is better), where the dashed line at Y=1 indicates performance equal to random initialization. The X-axis represents the training checkpoint index, starting from the random initialization at index 0. Lines: Show the normalized MSE for Mid (Post-Attention/Pre-MLP, blue circles) and Post (Post-MLP, orange squares) activations, reflecting their fit to their respective theoretical geometries over training. For Mid activations we regress to the Constrained Belief geometry, while for Post activations we regress to the Full (unconstrained) Belief geometry. Insets: Provide a zoomed-in view of the Y-axis from -0.02 to 0.5, highlighting the convergence behavior at low MSE values. **Observations:** Across all conditions, both Mid and Post activation representations show significantly improved geometric fits (normalized MSE drops well below 1) compared to the random baseline as training progresses. Some conditions (e.g., a=20) may show an initial transient increase in normalized MSE before rapid improvement.

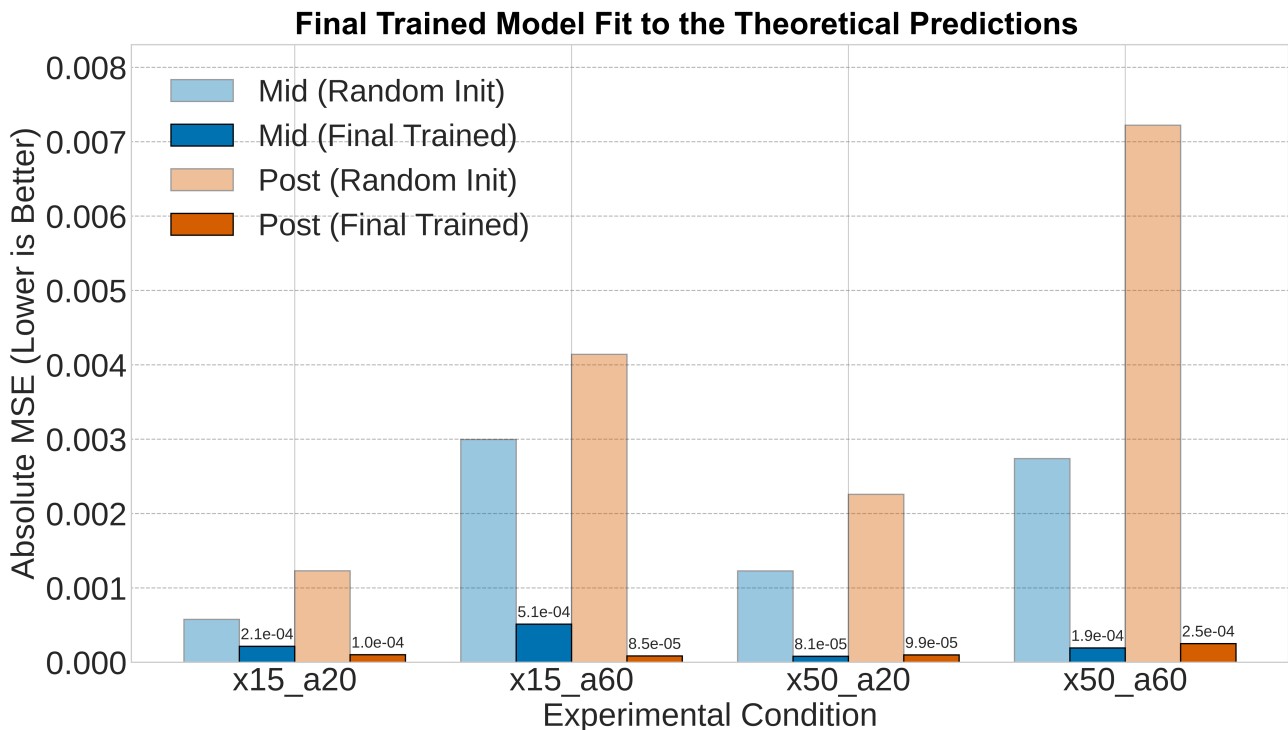

Figure A4: **Comparison of Final Trained Model Fit vs. Random Initialization Across Experimental Conditions.** The bar chart displays the Absolute Mean Squared Error (MSE, lower is better), comparing the geometric fit of activations at the final training checkpoint against the initial random model state. Results are shown for four different experimental conditions (X-axis). Within each condition, bars represent: Mid (Post-Attention/Pre-MLP) Activations, MSE of the regression to the Constrained Belief Geometry: Blue bars. Post (Post-MLP) Activations, MSE of the regression to the Full (unconstrained) Belief Geometry: Orange bars. Solid bars indicate the MSE for the final trained model, while transparent bars of the same color show the MSE for the randomly initialized model (baseline). Numerical labels specify the precise MSE values achieved by the trained models. **Observations:** Across all conditions, the trained model (solid bars) achieves lower MSE, indicating a much better fit to the underlying theoretical geometries compared to the random baseline (transparent bars).

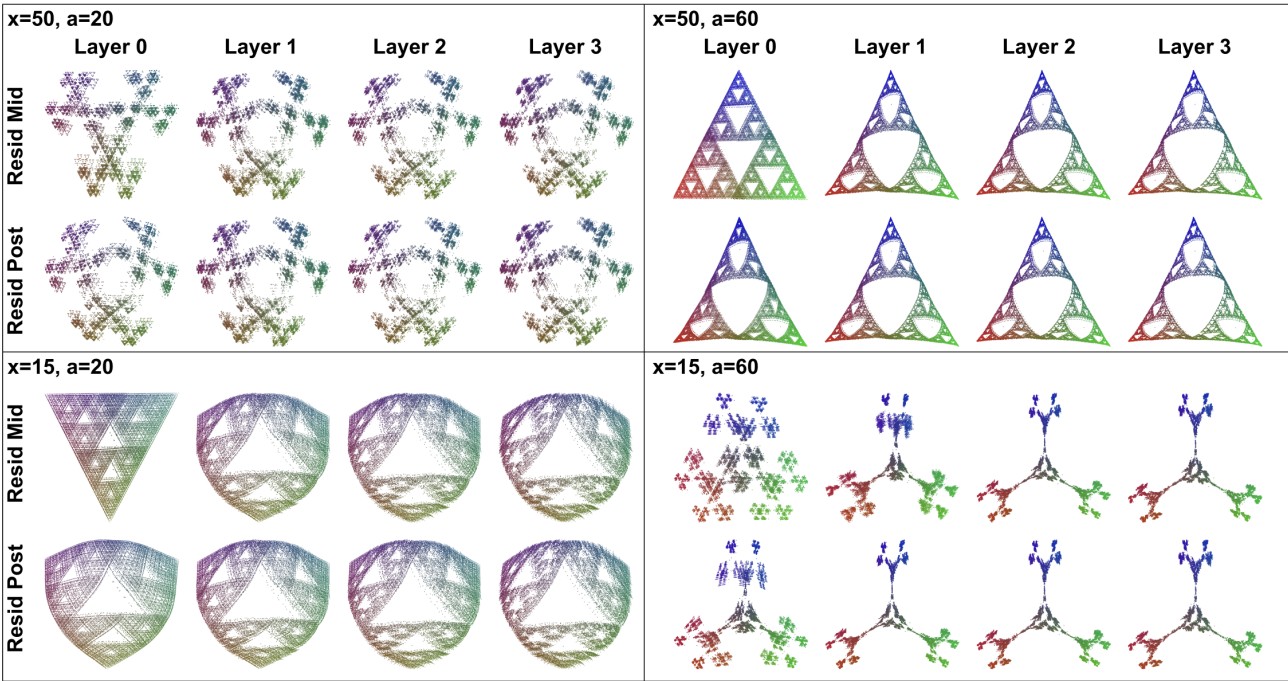

Figure A5: **Visualization of Activation Geometry via Regressions onto the Full Belief State.** The figure displays 2D projections of transformer activations after linear regression onto the Full (unconstrained) Belief state geometry (Eq. (2)) for all stages, arranged in a 2x2 grid. Each quadrant corresponds to a different experimental condition (combinations of x and a, labeled above each quadrant). Within each quadrant, the top row shows visualizations for Mid (Post-Attention/Pre-MLP) activations, and the bottom row shows visualizations for Post (Post-MLP) activations, across Layers 0-3 (columns). Points in all plots are colored according to the ground truth Full Belief state coordinates (Eq. (2)), projected into RGB space. Importantly, all visualizations show regressions to the Full Belief state geometry (and not the Constrained Belief geometry). Top Row (Mid Activations): Shows Mid activations after being regressed onto the Full Belief geometry (Eq. (2)). Bottom Row (Post Activations): Shows Post activations after being regressed onto the Full Belief geometry (Eq. (2)). This visualization qualitatively shows how well activations linearly map to the Full Belief state throughout the network. Notably, the residual stream activations after the first attention layer but before the MLP (Layer 0 mid, top-left plot within each quadrant), despite being regressed onto the Full Belief target, visually retain a structure resembling the Constrained Theory geometry (Eq. (5)). This indicates that the constrained structure is the dominant feature linearly recoverable from early Mid activations, even when seeking the best fit to the final target geometry. Comparing subsequent Mid and Post stages across layers (moving rightwards) reveals the accurate fit to the Full Belief state geometry.

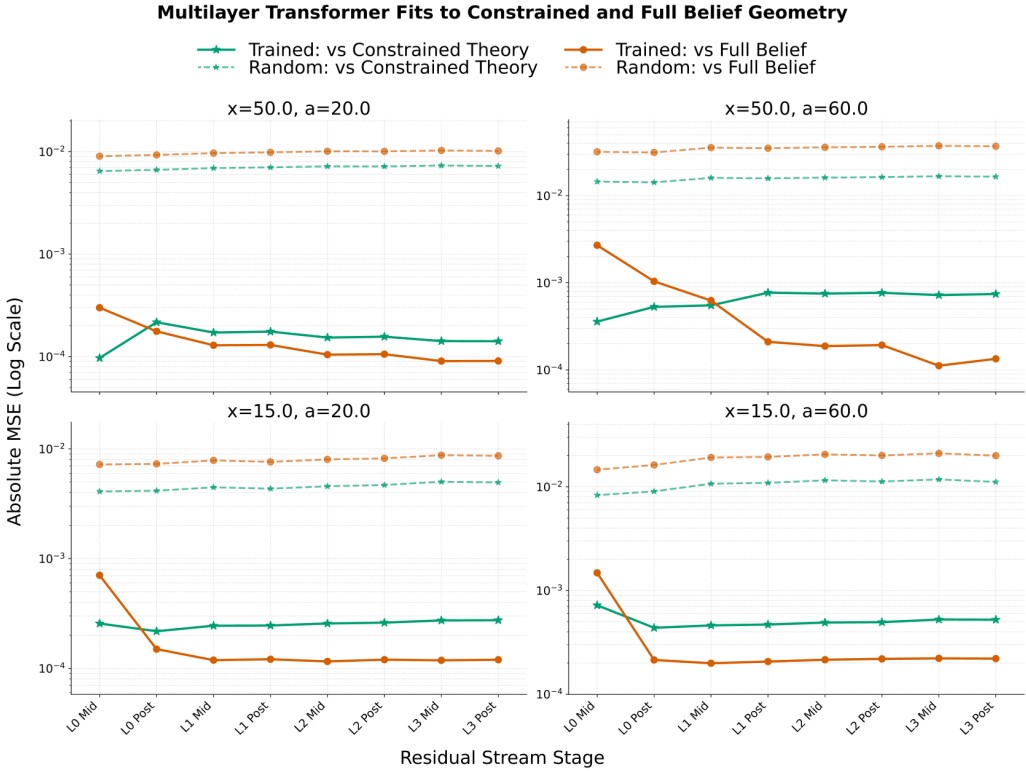

Figure A6: **Quantitative Fit of Multi-Layer Transformer Activations to Theoretical Geometries Across Residual Stream Stages.** Absolute Mean Squared Error (MSE, log scale) comparing activation representations to the Constrained Theory (Eq. (5)) and the Full (unconstrained) Belief state geometry (Eq. (2)) across interleaved residual stream stages (L0 Mid, L0 Post, ..., L3 Post) of a 4-layer transformer. The figure presents results for four experimental conditions (combinations of x=15, 50 and a=20, 60) in a 2x2 grid. Lines: Show absolute MSE comparing activation fits to theoretical geometries. Solid lines represent the trained model; dashed lines represent the random baseline. Green Stars: Fits of the residual stream activations to the Constrained Theory (Eq. (5)). Orange Circles: Fit of the residual stream activations to the Full Belief geometry (Eq. (2)). At the residual stream after the first attention but before the first MLP (L0 Mid), the fit to the Constrained Theory (green) is better (lower MSE) than the fit to the Full Belief geometry (orange) across all conditions, supporting the hypothesis that the initial layer's attention mechanism implements the constrained update. The fit to the Full Belief geometry (orange line) improves dramatically after the first MLP, and at that point the residual stream activations switch to fitting the Full Belief geometry better than the Constrained Belief geometry. The fit to the Constrained Theory (green line) does not show this convergence and may worsen in later layers. The MSE values for the trained transformer (solid lines) are consistently orders of magnitude lower than the corresponding random baselines (dashed lines), demonstrating that these geometric alignments are learned features resulting from training.

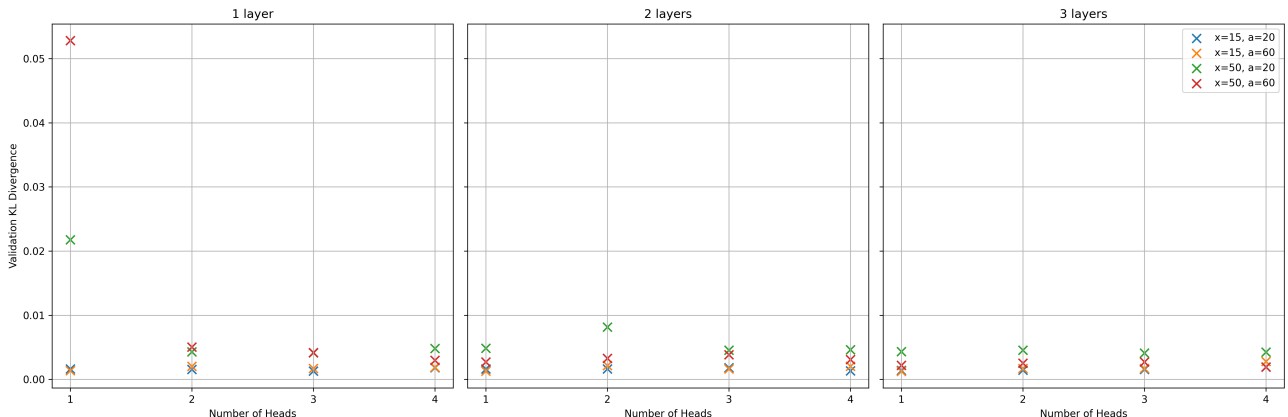

Figure A7: Validation KL divergence between model predictions and optimal probabilities across different architectural configurations. Results shown for various Mess3 parameter settings ($x$ and $\alpha$) and model architectures (number of heads and layers). The model achieves good performance with minimal architecture: a single layer with two attention heads is sufficient across parameter settings.

parameter settings. This empirical finding aligns with our theoretical analysis - when $x > 1/3$, the belief update patterns contain oscillatory components that require two heads to implement due to the non-negativity constraint of attention. The necessity of two heads is visually demonstrated in Fig. A8. For $x = 0.5$, where the optimal update pattern has significant oscillatory components, a single-head transformer fails to capture the correct belief geometry. With two heads, the model can properly implement these updates through complementary attention patterns, resulting in representations that closely match the ground truth geometry.

## G. Dimensionality of Residual Stream Activations

Table 1: Cumulative explained variance ratios for PCA components of the residual stream activations at the intermediate position (after attention) and the final position (before unembedding). The table shows results for different settings of the Mess3 HMM parameters $x$ and $\alpha$.

|  |  | Intermediate | | | | Final | | | |
|---|---|---|---|---|---|---|---|---|---|
|  | $x$ | 0.15 | 0.15 | 0.5 | 0.5 | 0.15 | 0.15 | 0.5 | 0.5 |
| component | $\alpha$ | 0.2 | 0.6 | 0.6 | 0.2 | 0.2 | 0.6 | 0.6 | 0.2 |
| 0 |  | 0.5408 | 0.4648 | 0.4074 | 0.5268 | 0.9618 | 0.4947 | 0.4596 | 0.6503 |
| 1 |  | 0.8768 | 0.8894 | 0.8028 | 0.8519 | 0.9825 | 0.7681 | 0.7096 | 0.8592 |
| 2 |  | 0.9673 | 0.9859 | 0.8913 | 0.9173 | 0.9943 | 0.9811 | 0.8855 | 0.9689 |
| 3 |  | 0.9749 | 0.9903 | 0.9455 | 0.9649 | 0.9960 | 0.9897 | 0.9189 | 0.9755 |
| 4 |  | 0.9815 | 0.9929 | 0.9848 | 0.9886 | 0.9969 | 0.9916 | 0.9428 | 0.9807 |
| 5 |  | 0.9870 | 0.9942 | 0.9978 | 0.9977 | 0.9976 | 0.9931 | 0.9586 | 0.9850 |
| 6 |  | 0.9914 | 0.9955 | 0.9986 | 0.9984 | 0.9981 | 0.9945 | 0.9723 | 0.9886 |

We perform PCA on the residual stream activations after the attention module (intermediate) and before the unembedding layer (final). The effective dimensionality of the residual stream is low, with the first few components capturing most of the variance (See Table 1). In most cases, the first 3 components explain over 90% of the variance. For $x = 0.5$, the effective dimensionality is higher, possibly due to the oscillatory dynamics of the belief updating equation in this regime. Further investigation is needed to fully understand this phenomenon.

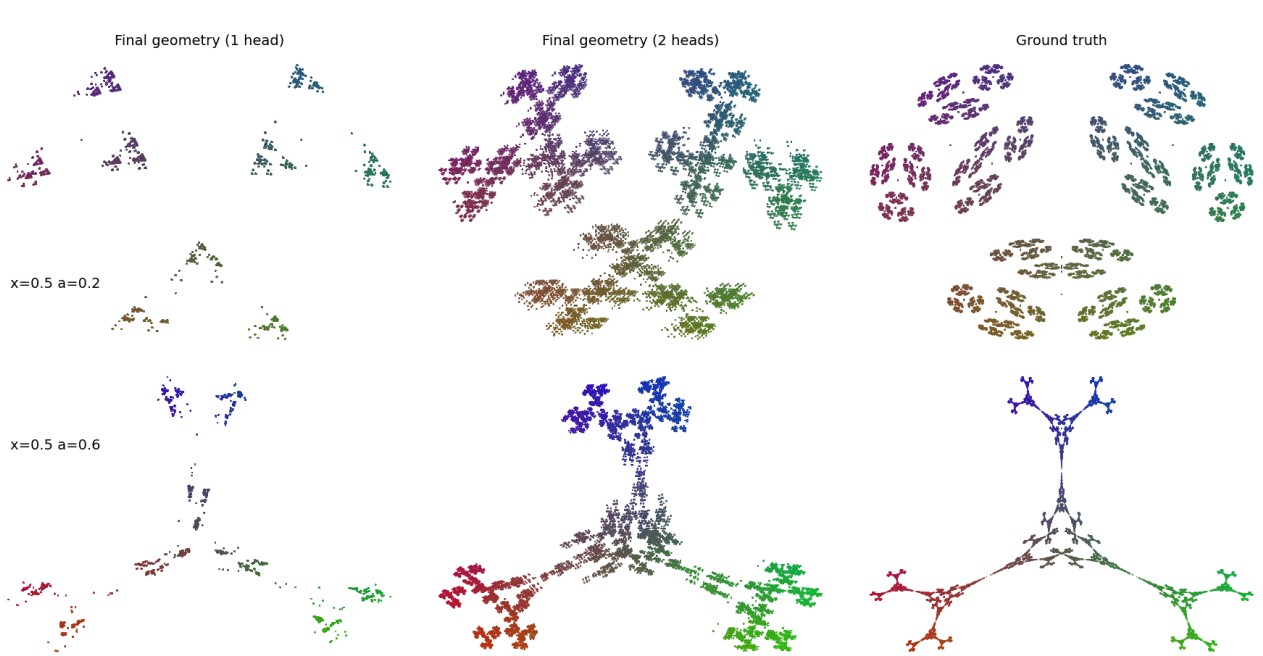

Figure A8: Comparison of learned belief geometry with one head (left) versus two heads (middle) against ground truth (right) for two different Mess3 parameter settings. With $x = 0.5$, where the optimal update pattern requires both positive and negative components, a single head fails to capture the correct geometry due to the non-negativity constraint of attention. Two heads allow the model to properly implement these updates, resulting in geometry that closely matches the ground truth.

