# OpenReview forum: "Constrained Belief Updates Explain Geometric Structures in Transformer Representations"
_ICML.cc/2025/Conference — ICML 2025 poster_

### Official Review · Reviewer_Wjfr · 2025-03-10

**Overall Recommendation:** 3

**Summary:**

In this paper, the authors propose a theoretical framework suggesting that transformers implement constrained Bayesian belief updating, and explain observed geometric patterns in transformer representations. Using the tools of mechanistic interpretability, they derive and empirically validate precise predictions of attention patterns, intermediate fractal-like structures, and final belief states in the trained transformers. The study focuses on the Mess3 family of Hidden Markov Models (HMMs).

## Update after rebuttal
I believe the authors have addressed and clarified some of my concerns. I maintain some reservations on the generalizability of the present findings, especially given the ad hoc nature of the considered learning task (not clearly connected to realistic tasks where transformers are employed) and of the theoretical analysis, here made possible because of the simplicity of the task. I also think that giving more prominence to the requisites for converging to a good representation (in terms of amount of data and training) could clarify when such a "constrained" regime could appear in the training of a transformer in a more realistic setting.
However, I believe the paper is well written, and the presented analysis is sound and original, so I will raise my score to a 3.

**Claims And Evidence:**

The paper makes clear theoretical claims regarding the correspondence between transformer mechanisms and Bayesian belief updating constrained by architectural elements. Empirical evidence from analyses of transformers trained on synthetic Mess3 datasets mostly validates these claims, although some discrepancies (e.g., in the analysis in the appendix, and when the alpha parameter exceeds a certain range) are not addressed in the present work, and in my opinion are slightly at odds with the strong claim of full interpretability of the model computations.

On the other hand, in the abstract the authors state that their approach "provides a principled lens on how gradient descent resolves the tension between optimal prediction and architectural design", but in the paper there is no analysis of the learning procedure and the "algorithm discovery" phase, since the analysis only takes into account the asymptotic solution after training (and does not even consider what happens in a data scarse regime).

**Essential References Not Discussed:**

None.

**Experimental Designs Or Analyses:**

The empirical evaluations, confined exclusively to synthetic datasets (Mess3), mostly validate the theoretical predictions of the authors. However, the authors hint at the possible discrepancies that can arise in some regimes that are not investigated in this paper (e.g., large alpha) and some predictions that are not matched by the transformer implementation are left without explanation. This undermines the claims about the explanatory framework presented by the authors.

**Methods And Evaluation Criteria:**

The authors utilize mechanistic interpretability (analysis of attention circuits and residual streams) and principal component analysis (PCA) to demonstrate the emergence of predicted geometric structures within the transformers' internal representations, relying on the synthetic and tightly controlled scenario under study. However, in some cases the authors rely on visual evaluation for quantifying the degree of agreement between their predictions and the empirical results (see e.g. fig. 4, where there is a general qualitative agreement but also visible differences, and assessing how relevant they are is difficult for the reader).

**Other Comments Or Suggestions:**

The authors should explore practical implications or architectural modifications inspired by their theory. Additionally, discussing the conditions under which the theory may fail or require adjustments in more realistic or complex tasks would provide valuable balance and depth.

**Other Strengths And Weaknesses:**

*Strengths:*
* Rigorous mathematical derivations with clear empirical support in a controlled environment.

*Weaknesses:*
* Empirical scope confined to toy synthetic scenarios.
* Limited to no analysis of the deviations from the predicted computational behavior.
* No analysis of the training process and how the optimization lands on the discovery of the optimal inference algorithm.
* Limited discussion of how findings extend or generalize to practical transformer models on natural language data.

**Questions For Authors:**

* Can the authors discuss how gradient descent helps the transformer converge to the optimal representation? E.g., how much data is needed?
* Can the authors discuss when there are deviations from the predicted behavior and why? Does something change qualitatively in the inference model (e.g. at large alpha)?
* Is the MLP layer warping mechanism something the authors think will extend to different settings?
* How might the theoretical framework be extended to transformers trained on natural language data?

**Relation To Broader Scientific Literature:**

While the paper appropriately references computational mechanics and transformer interpretability literature, it does not contrast this study against other mechanistic or interpretability-focused frameworks (e.g., circuit-based methods, dictionary learning, causal interventions), making its novelty and unique advantages somewhat unclear. Many of these works also study how exact inference can be embedded within the architectural constraint of the transformer.

**Theoretical Claims:**

The presented theoretical framework is mathematically rigorous and predicts intermediate and final representation geometries. However, assumptions behind this theoretical setup, particularly regarding optimal Bayesian inference in constrained attention mechanisms, require clearer justifications, especially concerning their applicability to natural tasks beyond simplified HMM settings. In particular, the authors claim that it is crucial to investigate how model-specific constraints can warp the optimal Bayesian inference, but it is not clear how this interpretative route can be generalized to more complex cases (e.g. in LLMs trained on natural text, where there are open questions about the computational order mismatch between the transformer computations and the correct inference procedure).

---

> ### Author Rebuttal · Authors · 2025-04-01
>
> Thank you for your thorough review and insightful feedback. We appreciate your recognition of our works rigor and the empirical support in our controlled setting. We also acknowledge the validity of your concerns regarding quantitative evaluation, learning dynamics, scope, and unexplained deviations. We have performed new analyses and made new figures to address these points and will make changes to the text, described below.
>
> **1. Learning Dynamics & Quantitative Evaluation:** You rightly pointed out the lack of analysis of the learning procedure despite the abstract's phrasing, and the need for quantitative evaluation beyond visual inspection (e.g., Fig 4).
>
> * **Learning Dynamics:** We have tracked how well the network activations fit our theoretical predictions during training. New Figure S ([link](https://bit.ly/4ldbEVZ)) shows the Mean Squared Error (MSE) when regressing activations onto theoretical geometries (Constrained Belief for Post-Attention, Full Belief for Post-MLP) throughout training. It demonstrates that the model rapidly converges towards the predicted geometries (Normalized MSE \<\< 1), quantitatively showing *that* gradient descent finds these solutions.
> * **Quantitative Fit:** New Figure R ([link](https://bit.ly/3QTfFkG)) provides the absolute final MSE values, confirming a quantitative fit to our theory in the trained model compared to random initialization, addressing the need for metrics beyond visual assessment.
> * **Abstract Phrasing:** We agree the phrasing "gradient descent resolves the tension" is misleading. Our core point is how the transformer's architecture itself (parallel attention vs. recursive Bayes) necessitates a specific computational form (Constrained Belief Update) to approximate the optimal solution. While Fig S shows GD finds this solution, our analysis focuses on what solution the architecture predisposes. We will revise the abstract to emphasize the role of architectural constraints in shaping the emergent computation, and get rid of the reference to GD.
>
> **2. Scope (Single-Layer Focus & Generalizability):** Our core derivation indeed focuses on the computation within the first layer. However, we have now analyzed deeper models.
>
> * **Multi-Layer Analysis:** As detailed in our response to Reviewer tAc4, we analyzed 4-layer transformers. New Figures X (quantitative MSE across layers ([link](https://bit.ly/4iP8vdo)) and Y (qualitative PCA visuals across layers ([link](https://bit.ly/4ch5W1m)) show that the 1st attn. layer consistently implements the Constrained Belief update, while subsequent layers refine the representation towards the Full Bayesian belief state.
> * **HMM Focus & Generalization:** We used Mess3 HMMs not just for tractability, but because such controlled systems are essential for isolating and rigorously testing fundamental principles – like how architectural constraints shape optimal prediction – before scaling hypotheses to the complexities of natural language, where similar principles likely apply but are harder to isolate. As for the MLP, it's role in refining might generalize, though its specific function will be task-dependent.
>
> **3. Discrepancies & Justification:** We acknowledge the discrepancies noted (e.g., Appendix C embeddings, deviations outside the central alpha range mentioned in Sec 4.3).
>
> * **Contextualizing Deviations:** While these deviations warrant further study (as noted in our limitations), the new quantitative results (Figs R, S, X) demonstrate that the overall fit to our theory is strong. Also, at large alpha Mess3 generates near IID data, which leads to a lack of learnable structure - we believe this explains why there are deviations at large alpha.
> * **Quantitative Predictions:** Importantly, our spectral theory correctly predicts the necessity of multiple attention heads when the HMM transition matrix has negative eigenvalues (Sec 4.4.1, Fig 3), a non-trivial validation of the underlying principles.
> * **Justification for Eq. 5:** We have clarified the derivation of the Constrained Belief Update (Eq. 5) in our response to tAc4, including a new diagram [link](https://bit.ly/3XGCU5g) explaining why it represents the natural parallel approximation achievable by a single attention layer due to its architectural inability to access intermediate tokens recursively. This clarification hopefully makes the theoretical motivation and assumptions behind our core mechanism clearer.
>
> **4. Relation to Literature:** Our work's unique contribution is in using computational mechanics not just to describe optimal prediction, but to theoretically predict how specific architectural constraints (attention's parallel structure, preventing recursive updates) lead to systematic, geometric deviations (the Constrained Belief Update) from the unconstrained Bayesian ideal. This predictive, theory-driven approach complements empirical circuit discovery or other mech interp. methods.
>
> Thank you again for your valuable feedback!

---

### Official Review · Reviewer_tAc4 · 2025-03-13

**Overall Recommendation:** 4

**Summary:**

This paper studies internal representations in transformers trained with next-token prediction on sequences generated by Hidden Markov Models. It reveals that transformers (focusing on single-layer transformers) perform constrained Bayesian belief updates, implementing an approximate version of optimal Bayesian inference in their parallel architecture. The paper proposes a mechanistic algorithm allowing attention heads to implement this process, creating predictable geometrically structured representations. The authors also develop a spectral theory to explain how data-generating transition matrices influence attention behavior. Through experiments on Mess3 class Hidden Markov Models, the authors validate their predictions about attention patterns and representation geometry. In summary, this work investigates the connection between optimal prediction theory and mechanistic interpretability, providing a principled understanding of how transformers balance optimality and architectural design.

**Claims And Evidence:**

I find the empirical evidence in support of all claims made in this publication sufficiently compelling. The authors convincingly demonstrate that the proposed belief state update algorithm is the primary mechanism implemented by the single-layer transformer. A separate treatment of the negative eigenvalue case and the role of multi-head attention is also quite notable. My only concern at the moment is the observation of a "scalar discrepancy in the first two embedding vectors". I do not see this as being critical for the main conclusions of the paper, but hope that this might be understood eventually.

**Essential References Not Discussed:**

I am not aware of any essential references that were not discussed in the paper.

**Experimental Designs Or Analyses:**

I reviewed the overall design of experiments aimed at exploring the properties of trained transformers and comparing them with theoretical predictions. The proposed analysis seems adequate and appears to capture remarkable similarity between theoretical expectations and properties of trained models.

**Methods And Evaluation Criteria:**

Given the nature of this publication, the simplistic Mess3 class Hidden Markov Models appears to be an adequate choice that allowed the authors to probe the precise mechanisms implemented by the transformer model. The paper could potentially benefit from discussing more general families of HMMs and potential qualitative differences in the state dynamics, but in my opinion, this is not critically important.

**Other Comments Or Suggestions:**

My primary suggestion is to strengthen the paper's narrative. I believe it would benefit from more in-depth discussions, such as the derivation of equation (5) and highlighting potential state geometry differences in other HMMs. If space is limited, detailed derivations could be moved to the appendix.

**Other Strengths And Weaknesses:**

In my opinion, two most important strengths of this publication are: (a) its creative setup and approach to understanding how transformers gradually attain information about the hidden state underlying the observed system dynamics; (b) the depth of theoretical understanding of very particular mechanisms that a trained transformer appears to implement.

I also identified two weaknesses that made understanding this paper a little difficult. One is that the proposed mechanism and experimental studies seem to be focused on a single-layer transformer. The publication hardly mentions this fact and there appear to be just a few places (one in Appendix) where this exceptionally important detail is stated explicitly. This makes the main contributions stated by the authors in the abstract and the introduction less general than they may appear to be. I would expect the learned mechanisms in multi-layer transformers to be much more nuanced and while this clearly falls outside the scope of the current publication, it needs to be stated very explicitly.

Secondly, the narrative structure of the paper is a little difficult to follow at times. For example, an exceptionally important equation (5) is not explained sufficiently well. It is my understanding that it can be related to the lowest-order term in the decomposition of the operator product $(T+\epsilon T_1)(T+\epsilon T_2)\dots (T+\epsilon T_n) \approx T^n + \epsilon (T_1 T^{n-1} + T T_2 T^{n-2} + \dots + T^{n-1} T_n) + \dots$, but this needs to be explained. Otherwise, it is not entirely clear why this particular expression is “one and only” sensible way of approximating the constrained belief update using a single-layer attention mechanism.

**Questions For Authors:**

Two primary questions I have are related to two potential weaknesses:

1. Is my understanding correct and all stated results (both theoretical and experimental) are only in relation to single-layer transformers. If so, have authors considered a very natural multi-layer extension? What were the most notable differences in how multi-layer transformers learn the updates in their residual stream?

2. Is my understanding of the origins of equation (5) correct? If not, what is the explanation and are there any formal statements explaining why we expect this exact form to be implemented by a self-attention layer?

**Relation To Broader Scientific Literature:**

This work investigates the specific mechanisms by which transformers capture belief state geometry in their residual stream, building upon a setup similar to “Transformers Represent Belief State Geometry in their Residual Stream.” It contributes to the broader field of mechanistic interpretability of transformers, which includes studies on linear and nonlinear regression, arithmetic, and other tasks. The paper also relates to research exploring transformers' ability to learn Hidden Markov Models (HMMs), as well as studies on the limitations of transformers in this context (for example, “On Limitation of Transformer for Learning HMMs”). Finally, it connects to: (a) 'An Explanation of In-context Learning as Implicit Bayesian Inference,' which uses HMMs to understand in-context learning through Bayesian inference, and (b) research on the information-theoretic properties of in-context learning (the observed fractal-like structure of system attractors can be linked to the growth of mutual information between the observed token sequence and the current state representation).

**Theoretical Claims:**

While I have not exhaustively verified all of the claims (for instance, I have not tried computing the eigenvalues of T), I read all of the authors theoretical claims and could not identify any clear mistakes. Most theoretical statements and intermediate results appear to be correct.

---

> ### Author Rebuttal · Authors · 2025-04-01
>
> Thank you for the positive assessment of our work, and also your constructive feedback, which improved the clarity and impact of our work.
> We address your primary concerns regarding the single-layer focus and the justification for Eq. 5, as described below, with new figs and analysis:
>
> **Single Layer Focus and Multi-Layer Extension** You are correct that our analysis focuses on a single-layer transformer, and we acknowledge this could have been stated more explicitly throughout the main text. We will revise the Abstract, Introduction, Methodology, and Discussion sections to clearly delineate that the primary analysis investigates the computation within the first layer, while also discussing how this relates to deeper networks. Although we decided to focus on the first layer for this initial work, we plan on extending the theory out to the multilayer setting in future work.
>
> To directly address your question about multi-layer extensions in experiment, we have trained and analyzed 4-layer transformers. Our key finding is that the first attention layer consistently implements the constrained belief update mechanism (Eq. 5) derived in the paper, producing the intermediate fractal geometry. Subsequent layers (specifically the MLPs and later attention blocks) then act to transform this intermediate representation progressively closer to the full Bayesian belief state geometry (Eq. 2).
>
>  We will add quantitative evidence for this to the appendix the main text. This includes:
>
> - PCA visualizations of activations after each layer in the 4-layer models, showing the transition from constrained to full belief geometry. New fig [here](https://bit.ly/4ch5W1m)
> - Mean Squared Error (MSE) analysis from linear regressions of residual stream activations onto both the theoretical constrained geometry (Eq.5) and the full belief geometry (Eq.2). This analysis, performed layer-by-layer, quantitatively shows the initial fit to Eq.5 post-attention-1, and the increasing fit to Eq.2 in deeper layers. New fig [here](https://bit.ly/4iP8vdo)
>
> These results demonstrate that our single-layer analysis captures the fundamental computation performed by the initial attention mechanism, even within deeper networks, while later layers perform refinement.
>
> **Intuition for Eq.5:** We appreciate you highlighting the need for a clearer explanation of Eq. 5. We will incorporate the following improved intuition into Section 4.3:
> - Bayesian inference (Eq. 2) requires multiplying token-specific transition matrices, a fundamentally recursive process where updates depend on the full history integrated up to the previous step.
> - In contrast, an attention head computes its output at position d via a parallel, feedforward weighted sum (Eq. 3) of value vectors ($v_s$) from source positions $s \le d$.
> - Crucially, each $v_s$ contains only local information from position $s$. It cannot directly access or depend on the specific tokens between $s$ and $d$ due to the parallel nature of the value computation and attention weighting.
> - Therefore, the most information that token z_s can independently contribute to the belief at position d within this single-layer constraint is the correction derived from knowing $z_s$ occurred d-s steps prior, assuming a default starting belief ($\pi$) and no knowledge of intervening tokens. These independent displacements from the stationary distribution over latent states is the difference of probability distributions: $Pr(S_d | Z_s) - Pr(S_d)$. Linear algebraically, this contribution is precisely $\pi T^{|z_s} T^{d-s} - \pi$.
> - Summing these over all past sources naturally yields the constrained belief update form in Eq. 5. It represents the best possible parallel approximation achievable by a single attention layer given its architectural limitations.
> - [link](https://bit.ly/3XGCU5g) depicts each of these above points.
> We believe this revised explanation clarifies why Eq. (5) is the natural computation for the attention layer to implement when reconciling optimal prediction with its inherent structure.
>
> Regarding the minor scalar discrepancy for the first two embeddings, we agree it doesn't invalidate the core findings. While still under investigation (as noted in Appendix C), the remarkable agreement across the attention pattern, OV vectors, and subsequent embeddings strongly supports our theoretical framework.  And as the referee noted, the spectral perspective allows us to anticipate that multiple attention heads are needed when the transition matrix for the training data has negative eigenvalues.
>
> We will also strengthen the narrative as suggested, ensuring the scope (initial focus on first layer) and the derivation logic are clearer. While exploring other HMMs is interesting future work, Mess3's tractability was essential for this detailed mechanistic study.
> We hope these clarifications and planned revisions address your concerns and demonstrate the significance of our findings. Thank you again for your valuable feedback.

---

> > ### Comment · Reviewer_tAc4 · 2025-04-04
> >
> > I would like to thank the authors for their detailed reply. I do believe that the proposed modifications will make the publication much clearer. While my understanding of the derivation appears to have been correct, I am glad that it will be stated much more explicitly in the main text. Additional empirical multi-layer results are also quite interesting and insightful.

---

### Official Review · Reviewer_bbxf · 2025-03-13

**Overall Recommendation:** 4

**Summary:**

This work bridges mechanistic interpretability and Bayesian definitions of optimal predictions (constrained belief updating) to study emerging geometric structures in small (1-layer) transformer models. This is done with a solid theoretical foundation, in a ‘controlled’ experiment: We know the generative model behind the dataset used for training (the Mess3), that is an HMM with 3 states, that depends on two parameters.

Sadly, this is quite far from my area of expertise, so please do not give much weight to my review, and forgive misunderstanding/mistakes.

**Claims And Evidence:**

The claims seem well supported by the methodology. In this, it is a good work: research question, claim, method,  and experimental evidence are clearly explained.

**Essential References Not Discussed:**

I'm not familiar with the literature.

**Experimental Designs Or Analyses:**

They seem valid, the models used are justified.

**Methods And Evaluation Criteria:**

The proposed method and evaluations are well motivated and defined, in a controlled setup.

**Other Comments Or Suggestions:**

Minor: Z_{d+1} for a sequence is confusing, maybe use Z_{d:T}, or some other notation to express that it is a series and not a single token?

**Other Strengths And Weaknesses:**

**Pros**: The paper seems very well motivated and structured, and the fact that it uses a ’toy’ controlled setup for training and testing makes it more reliable, and allows to produce testable predictions. I have also enjoyed the discussion on one and two head transformer models. For what I could understand, it was quite an interesting read.


**Cons**: I found some parts of the paper, especially the parts related to the spectral algorithm harder to understand. While this is not a cons overall (it could be given by my ignorance on the topic), I would suggest the authors to try to add a couple more intuitive explanations.

Also, I think it is not clear, how the fractal drawings emerges from the belief states. As it is a key point, maybe it could be worth to add a couple more lines of explanation to this, either in the main body, or in the first section of the supplementary material.

**Questions For Authors:**

No questions

**Relation To Broader Scientific Literature:**

They allow us to understand transformer models a little better than before.

**Theoretical Claims:**

No, they are outside my area of expertise

---

> ### Author Rebuttal · Authors · 2025-04-01
>
> Dear Reviewer bbxf,
>
> Thank you for your positive assessment and thoughtful feedback. We appreciate that you find the work well-motivated and structured. We care very much about making the paper understandable and clear, so your suggestions for clarification are very helpful!
>
> **Emergence of Fractal Geometry:**
>
> You asked for clarification on how the fractal drawings emerge from the belief states. That's an excellent point. These structures (visualized in Figs 1D, 4) arise directly from the iterative application of Bayesian belief updating rules within the probability simplex representing beliefs over the HMM's hidden states.
>
> - Each time a new token is observed, the current belief state (a point in the simplex) is transformed according to the update equations (Eq. 2 for the full Bayesian update, Eq. 5 for our derived constrained approximation implemented by attention).
> - Because the underlying Mess3 HMM involves probabilistic transitions between states, applying these update rules repeatedly for different token sequences acts mathematically like an Iterated Function System (IFS). IFSs are known to generate self-similar, fractal geometries.
> - The specific fractal shapes shown in our figures represent the collection of all possible belief states reachable after observing sequences up to a certain length. The "folding" or self-similarity comes from the fact that different sequences can lead to nearby belief states, and the update rules map regions of the simplex onto smaller regions within itself. (This connection between HMMs, Bayesian updates, and fractals is explored theoretically, e.g., by Jurgens & Crutchfield, 2021).
>
> We will add a clearer explanation of this link between the iterative belief updates and the resulting fractal geometry either in the main text (likely near Figure 1/Section 3, space permitting) and/or expand on it in Appendix A.
>
> **Intuition for the Spectral Algorithm (Section 4.3):**
>
> Thank you for highlighting that Section 4.3 could benefit from more intuition, which we will add to the text. The main goal of the spectral analysis is to understand precisely how information or influence from a past token (at source position s) propagates to affect the belief state at the current destination position d.
> - This influence mathematically depends on the sequence of hidden state transitions between s and d, captured by powers of the HMM transition matrix, $T^{d-s}$.
> - Spectral decomposition (using eigenvalues $\lambda$ and associated projectors $T_\lambda$) is a standard mathematical tool to analyze matrix powers because it simplifies $T^n$ into a sum $\sum \lambda^n T_\lambda$.
> - The eigenvalues ($\lambda$) are crucial because they tell us the rate at which the influence of past information decays (if $|\lambda|<1$) or even oscillates (if $\lambda$ is negative or complex) as the distance n=d-s increases.
> - Our key finding here is that the learned attention weights ($A_{d,s}$ in Eq. 3) directly implement this propagation effect, effectively learning to approximate the $\lambda^{d-s}$ decay predicted by the theory (Eq. 10). This explains why attention patterns often show exponential decay, and why multiple heads are needed (Fig 3) to capture oscillatory patterns arising from negative eigenvalues.
> - Furthermore, this spectral perspective allows us to make precise, verifiable predictions about the learned OV vectors ($\vec{v}_s$ in Eq. 13) and token embeddings ($\vec{x}_s$ in Eq. 14), directly connecting the dynamics of the data (via T's eigenvalues) to the specific parameters learned by the transformer.
>
> We will revise Section 4.3 to include a paragraph explaining this intuition more clearly.
>
> **Notation Suggestion:**
>
> We completely agree that using $Z_{d+1}:$ for a sequence could be confusing. Thank you for pointing this out! We will revise the notation throughout the paper to use $Z_{d+1:\infty}$ to unambiguously denote the sequence of future tokens.
>
> **Conceptual Diagram:**
>
> To further aid in understanding the core theoretical idea, we have also created a new diagram [link here](https://docs.google.com/document/d/1CDwiJp8qG4d4OqgntBkIMYQhxJ1vy-2j2BecubhrL5U/edit?tab=t.0#heading=h.pwspurd6saek), that visually explains how the architectural constraints of attention lead naturally from the ideal Bayesian update (Eq. 2) to our derived constrained belief update (Eq. 5).
>
> Thank you again for your helpful suggestions, which will undoubtedly improve the clarity of our paper. We appreciate your support for our work.
>
> Sincerely,
> The Authors

---

### Official Review · Reviewer_rrWD · 2025-03-16

**Overall Recommendation:** 5

**Summary:**

This paper discussed the circuit implemented within trained Transformer to implement (partial) Bayesian inference on a type of Hidden Markov Model. The paper combines theoretical analysis with interpretability tool to showcase that the implemented circuit and corresponding representation is optimal when accounting the architectural constraint of 1-layer Attention. Overall, this paper produce a satisfactory explaination on the inner working mechanism of Transformer on the Mess-3 HMM.

**Claims And Evidence:**

Yes, the claims are well supported.

**Essential References Not Discussed:**

No that I can think of.

**Experimental Designs Or Analyses:**

Yes.

**Methods And Evaluation Criteria:**

Yes.

**Other Comments Or Suggestions:**

In Section 2.2, it would be better to mention that the same deduction and notation is inherited from Shai et.al.

**Other Strengths And Weaknesses:**

# Strength
1. The visualization in Figure 2 is very intuitive.
2. The theory in Section 4 is clean and captures the essence of the architectural constraint.

# Weakness

1. Only 1 layer case is studied here, while this keeps the paper compact, how the current result generalize to multi-layer is unclear.

**Questions For Authors:**

1. Following weakness 1, has the authors tried to ablate with number of layer and will the first layer implement similar algorithm? Is there any new phenomenon in the multi-layer case that can't be explained by the 1-layer analysis here?

**Relation To Broader Scientific Literature:**

The authors made a very thorough discussion in Section 2.1 already. This paper should be of interest to the community of both mechanistic interpretability and representation theory.

The most related work is [1], which discuss the same setting and find that Transformers approximately learn to encode the belief state linearly in the hidden state. This paper produces a comprehensive analysis on how this belief state is calculated within Transformer in a 1-layer setting.


[1] Transformers represent belief state geometry in their residual stream

**Theoretical Claims:**

I checked the theoretical claims produced by this paper and am convinced that they are correct.

---

> ### Author Rebuttal · Authors · 2025-04-01
>
> Thank you for your strong endorsement of our work. We appreciate your positive assessment of our paper's strengths, and we agree that this paper should make a nice contribution to both interpretability and the study of representations.
> We also appreciate your constructive feedback, which nudged us to investigate and confirm that our results indeed generalize to the multi-layer case.
>
> **Multi-Layer Extension:**  To directly address your question about multi-layer extensions, we have trained and analyzed 4-layer transformers. Our key finding is that the first attention layer consistently implements the constrained belief update mechanism (Eq. 5) derived in the paper, producing the intermediate fractal geometry. Subsequent layers (specifically the MLPs and later attention blocks) then act to transform this intermediate representation progressively closer to the full Bayesian belief state geometry (Eq. 2).
>
>  We plan to add quantitative evidence for this to the appendix (and summarize in the main text). This includes:
>
>
> 1. PCA visualizations of activations after each layer in the 4-layer models, showing the transition from constrained to full belief geometry. See new figure [here](https://docs.google.com/document/d/1CDwiJp8qG4d4OqgntBkIMYQhxJ1vy-2j2BecubhrL5U/edit?tab=t.0#heading=h.7gjcfnihl09i)
> 2. Mean Squared Error (MSE) analysis from linear regressions of residual stream activations onto both the theoretical constrained geometry (Eq. 5) and the full belief geometry (Eq. 2). This analysis, performed layer-by-layer, quantitatively shows the initial fit to Eq. 5 post-attention-1, and the increasing fit to Eq. 2 in deeper layers. See new figure  [here](https://docs.google.com/document/d/1CDwiJp8qG4d4OqgntBkIMYQhxJ1vy-2j2BecubhrL5U/edit?tab=t.0#heading=h.uauqgkf7zz7l)
>
> These results demonstrate that our single-layer analysis captures the fundamental computation performed by the initial attention mechanism, even within deeper networks, while later layers perform refinement.

---

### Decision · Program_Chairs · 2025-05-01

**Decision:**

Accept (poster)

**Comment:**

This paper presents a well-executed and theoretically grounded analysis of how single-layer transformers trained on synthetic HMM data implement a constrained form of Bayesian belief updating. Using tools from mechanistic interpretability and computational mechanics, the authors derive and empirically validate predictions about attention patterns and geometric structures in internal representations.

The majority of the reviewers support acceptance from the beginning, citing strong technical contributions and a clear theoretical-experimental link. Some concerns were raised about scope and clarity, which were addressed through expanded analysis (including multi-layer extensions) and clearer explanations.

There was a strong concern about generalisability and missing training dynamics, but the rebuttal resolved it and helped prompt the work’s rigour and contribution.

While the synthetic setting limits immediate applicability, the paper makes a novel and substantive contribution to mechanistic interpretability by offering a predictive theory of transformer computation under architectural constraints. I recommend acceptance.